# Future–Past Visitation Redundancy for Rapid Coverage Exploration in Reinforcement Learning

## Abstract

In reinforcement learning, novelty-based exploration methods encourage agents to explore novel states, but do not explicitly aim to improve state space coverage efficiency, which is critical for efficient credit assignment and rapid adaptation to non-stationary rewards. To simultaneously consider state novelty and rapid coverage, we propose Future–Past Visitation Redundancy (FPVR) for exploration, induced by a Dual-Timescale State Representation (DTSR). DTSR maintains a successor representation learned over long timescales to encode the expected discounted accumulation of future state features, together with a persistence representation that records an online discounted accumulation of past state features. Exploration is guided by an FPVR-based bias on the $Q$ function, which suppresses short-term repetitive visitation. Experimental results in *MiniGrid* and *Atari 2600* games demonstrate that the proposed method effectively reduces state-space cover time and improves learning efficiency in sparse and non-stationary reward environments.

## 1. Introduction

Exploration remains a central challenge in reinforcement learning (RL), especially in sparse-reward and non-stationary settings where informative feedback is rare or drifting (Yang et al., 2021; Ladosz et al., 2022; Hao et al., 2023). Effective exploration should (i) seek novelty (Bellemare et al., 2016) to discover new states and sparse rewards and (ii) achieve rapid state-space coverage by reducing cover time (Amin et al., 2021a;b), which accelerates reward propagation/credit assignment and enables quick re-adaptation when the reward structure changes.

---

[1]Anonymous Institution, Anonymous City, Anonymous Region, Anonymous Country. Correspondence to: Anonymous Author <anon.email@domain.com>.

Preliminary work. Under review by the International Conference on Machine Learning (ICML). Do not distribute.

A large body of work operationalizes novelty using intrinsic rewards, including (pseudo-)count bonuses (Bellemare et al., 2016; Ostrovski et al., 2017; Lobel et al., 2023), prediction-error curiosity (Pathak et al., 2017; Jarrett et al., 2023), random-network prediction residuals (Burda et al., 2019; Yang et al., 2024), and episodic memory (Savinov et al., 2019; Badia et al., 2020; Henaff et al., 2022). While effective in many sparse-reward domains, these approaches typically optimize short-horizon novelty rather than an explicit objective for coverage efficiency; as a result, an agent may explore a locally novel region for an extended period. Moreover, intrinsic bonuses may decay as states become familiar (Jarrett et al., 2023; Castanyer et al., 2024), which can weaken re-coverage signals in non-stationary environments.

Coverage-oriented exploration has also been studied via temporally extended actions in hierarchical RL, e.g., option discovery from transition-graph topology (Machado et al., 2017; 2018b; Klissarov & Machado, 2023) and covering options that explicitly reduce expected cover time (Jinnai et al., 2019; 2020; Chen et al., 2022). However, many approaches require substantial early exploration to learn suitable transition-aware representations and/or maintain a non-trivial option set with initiation/termination machinery, leading to delayed gains or additional optimization overhead (Campos et al., 2021; Hutsebaut-Buysse et al., 2022; Bagaria et al., 2023).

To address these limitations, we propose *Future–Past Visitation Redundancy* (FPVR) for exploration, induced by a *Dual-Timescale State Representation* (DTSR). The key idea is to suppress short-term revisitation, which improves coverage while biasing exploration toward novel states. DTSR combines (i) a successor representation (SR) that predicts expected discounted future feature occupancy and (ii) a persistence representation (PR) that summarizes an online discounted accumulation of past state features. FPVR measures the overlap between SR and PR and is injected as a bias into the $Q$ function to discourage redundant transitions. For high-dimensional and continuous domains, we incorporate online feature whitening and compute FPVR as cosine similarity between SR and PR in the whitened space, yielding a scale- and correlation-normalized overlap metric under function approximation.

FPVR offers two key advantages. First, it reconciles the preference for novel states emphasized by novelty-based methods with the coverage-efficiency objective highlighted in hierarchical exploration. Second, FPVR is introduced as a bias on the $Q$ function, which avoids shaping and optimizing a non-stationary novelty-based intrinsic objective, and it does not require training or storing multiple temporally extended subpolicies. We integrate FPVR with DQN (Mnih et al., 2015) and evaluate it on *MiniGrid* and *Atari 2600* games featuring sparse and non-stationary rewards. The results demonstrate that FPVR effectively improves learning and coverage performance, outperforming several classical novelty-based exploration baselines.

## 2. Preliminaries

**Notation.** Uppercase (e.g., $S_t, A_t$) denotes random variables while lowercase (e.g., $s_t, a_t$) denotes their realizations. The superscript $(t)$ marks time-dependent functions defined using information available at time $t$, e.g., $\mathcal{F}^{(t)}(s_t, a)$. $\tilde{\cdot}$ indicates quantities in the whitened feature space, e.g., $\tilde{\phi}^{(t)}(s)$.

**Markov Decision Process and Action-Value Function.** We consider a Markov decision process (MDP) defined by the tuple $\mathcal{M} = (\mathcal{S}, \mathcal{A}, P, r, \gamma)$, where $\mathcal{S}$ and $\mathcal{A}$ denote the state and action spaces, $P(\cdot \mid s, a)$ is the transition kernel, $r(s, a)$ is the reward function, and $\gamma \in (0, 1)$ is the discount factor (Puterman, 2014; Sutton et al., 1998). For a policy $\pi(a \mid s)$, the action-value function is defined as

$$Q^\pi(s, a) = \mathbb{E}_\pi\left[\sum_{t=0}^\infty \gamma^t r(S_t, A_t) \,\middle|\, S_0 = s, A_0 = a\right], \tag{1}$$

where $S_{t+1} \sim P(\cdot \mid S_t, A_t)$ and $A_t \sim \pi(\cdot \mid S_t)$ for $t \geq 0$. The $Q$ function satisfies the Bellman expectation equation

$$Q^\pi(s, a) = r(s, a) + \gamma\, \mathbb{E}_{s' \sim P(\cdot|s,a),\, a' \sim \pi(\cdot|s')}\left[Q^\pi(s', a')\right]. \tag{2}$$

In value-based RL, actions are typically selected by (approximately) maximizing an estimate of $Q(s, a)$, e.g., via $\epsilon$-greedy or Boltzmann exploration (Sutton et al., 1998). In this work, exploration is introduced as an explicit bias on the $Q$ function, and we later instantiate this idea in a DQN-style framework (Mnih et al., 2015).

**Successor Representation.** The successor representation (SR) predicts discounted future state occupancy under a policy. For a fixed policy $\pi$, the action-conditioned SR is

$$M^\pi(s, a, s') = \mathbb{E}_\pi\left[\sum_{t=0}^\infty \gamma^t\, \mathbb{I}\{S_t = s'\} \,\middle|\, S_0 = s, A_0 = a\right], \tag{3}$$

where $s' \in \mathcal{S}$ and $\mathbb{I}\{\cdot\}$ is the indicator function (Dayan, 1993; Stachenfeld et al., 2017).

Given a transition $(s_t, a_t, s_{t+1})$ and a bootstrap action $\bar{a}_{t+1} \in \mathcal{A}$, a TD(0) update for each $s' \in \mathcal{S}$ is

$$\hat{M}(s_t, a_t, s') \leftarrow \hat{M}(s_t, a_t, s') + \eta\Big(\mathbb{I}\{s_t = s'\}$$
$$+ \gamma\, \hat{M}(s_{t+1}, \bar{a}_{t+1}, s') - \hat{M}(s_t, a_t, s')\Big), \tag{4}$$

with step size $\eta > 0$. Let $\hat{\mathbf{m}}(s_t, a_t)$ stack $\hat{M}(s_t, a_t, s')$ over $s'$ and let $\mathbf{e}_{s_t}$ be the one-hot basis of $s_t$. Then $\hat{\mathbf{m}}(s_t, a_t) \leftarrow \hat{\mathbf{m}}(s_t, a_t) + \eta\Big(\mathbf{e}_{s_t} + \gamma\, \hat{\mathbf{m}}(s_{t+1}, \bar{a}_{t+1}) - \hat{\mathbf{m}}(s_t, a_t)\Big)$.

The bootstrap action can be chosen on-policy as $\bar{a}_{t+1} \sim \pi(\cdot \mid s_{t+1})$, or off-policy via a greedy backup induced by an auxiliary action-value function $Q_0$:

$$\bar{a}_{t+1} \in \arg\max_{a \in \mathcal{A}} Q_0(s_{t+1}, a), \tag{5}$$

where $Q_0$ may be any action-value estimate used to define greedy backups (Kulkarni et al., 2016; Barreto et al., 2018). If the reward depends only on the visited state, i.e., $r_t = r(s_t)$, then

$$Q^\pi(s, a) = \sum_{s' \in \mathcal{S}} M^\pi(s, a, s')\, r(s'). \tag{6}$$

**Successor Features.** Successor features (SF) generalize SR to arbitrary features $\phi(s) \in \mathbb{R}^d$ (Barreto et al., 2017). For a policy $\pi$,

$$\psi^\pi(s, a) = \mathbb{E}_\pi\left[\sum_{t=0}^\infty \gamma^t \phi(S_t) \,\middle|\, S_0 = s, A_0 = a\right]. \tag{7}$$

Under the linear reward model $r(s) = w^\top \phi(s)$,

$$Q^\pi(s, a) = w^\top \psi^\pi(s, a). \tag{8}$$

Given a transition $(s_t, a_t, s_{t+1})$ and a bootstrap action $\bar{a}_{t+1} \in \mathcal{A}$, a one-step TD target is

$$y^\psi = \phi(s_t) + \gamma\, \psi_\theta(s_{t+1}, \bar{a}_{t+1}), \tag{9}$$

and $\psi_\theta$ is updated by minimizing the squared TD error, e.g.,

$$\theta \leftarrow \theta - \eta\, \nabla_\theta \big\| \psi_\theta(s_t, a_t) - y^\psi \big\|_2^2. \tag{10}$$

**Zero-phase Component Analysis (ZCA) Whitening.** Whitening normalizes feature scale and removes linear correlations. Given mean $\mu_t$ and covariance $\Sigma_t$ of $\phi(s)$, ZCA whitening applies

$$\tilde{\phi}^{(t)}(s) = (\Sigma_t + \varepsilon I)^{-1/2}\big(\phi(s) - \mu_t\big), \tag{11}$$

where $\varepsilon > 0$ is a small regularizer. The resulting features have approximately identity covariance. Among whitening transforms, ZCA preserves maximal alignment with the original coordinates and minimizes $\ell_2$ distortion (Kessy et al., 2018), and is often more stable than PCA whitening under online estimates of $(\mu_t, \Sigma_t)$ (Huang et al., 2018).

# 3. Future-Past Visitation Redundancy for Exploration

FPVR is motivated by the observation that large overlap between predicted future visitation and recent visitation indicates redundant exploration. As illustrated in Figure 1, we penalize this redundancy to improve coverage efficiency. We next specify the representations for future and past visitations and how we measure their overlap.

## 3.1. Dual-Timescale State Representation

We develop a dual-timescale state representation (DTSR) of visitation that summarizes complementary future and past information. In this subsection we focus on the tabular setting with a finite state space $\mathcal{S}$.

**Representation for future visitation.** To characterize future visitation under a policy $\pi$, we use the action-conditioned successor representation. As in (3), $M^\pi(s, a, \cdot)$ predicts expected discounted future state occupancy, providing a long-timescale summary of future visitation patterns.

**Representation for past visitation.** Complementary to SR, we define a representation of recent past state occupancy as follows.

**Definition 3.1** (Persistence Representation). Let $\tau = (s_0, s_1, \dots)$ be a state trajectory generated by a policy interacting with the MDP $\mathcal{M} = (\mathcal{S}, \mathcal{A}, P, r, \gamma)$. For $\lambda \in (0, 1)$, the *persistence representation* (PR) is defined as

$$C_t(s) \doteq \sum_{k=0}^{t} \lambda^{t-k} \, \mathbb{I}\{s_k = s\}, \qquad s \in \mathcal{S}. \quad (12)$$

Equivalently, with $C_{-1}(s) \doteq 0$, PR admits the recursion

$$C_t(s) = \mathbb{I}\{s_t = s\} + \lambda \, C_{t-1}(s), \qquad s \in \mathcal{S}. \quad (13)$$

**Dual-timescale structure:** The SR and the PR constitute a pair of visitation summaries defined at distinct timescales. The SR encodes a *mathematical expectation* of discounted future state visit-counts under a policy, and therefore must be estimated from many state-transition trajectories through *long-timescale* sampling and bootstrapped updates, becoming relatively stable as the estimate converges. In contrast, the PR corresponds to a single realized trajectory and records a discounted snapshot of past state visitations, which can be computed efficiently over *short-timescale* and typically evolves rapidly as the trajectory unfolds. This distinction yields a dual-timescale view of state visitation: a slowly learned, expectation-based future summary and a rapidly updated, trajectory-specific past summary, which together underpin the following exploration mechanism.

## 3.2. FPVR in the Tabular Setting

In the tabular setting, both SR and PR admit a common linear-algebraic form. Let $\{\mathbf{e}_s\}_{s \in \mathcal{S}}$ denote the canonical one-hot basis of $\mathbb{R}^{|\mathcal{S}|}$ indexed by states. Any discounted visitation summary can be written as a nonnegative linear combination of these basis vectors. In particular, at time $t$, the SR and PR in vector form can be expressed as

$$\mathbf{m}(s_t, a_t) = \sum_{s \in \mathcal{S}} a_s \, \mathbf{e}_s, \qquad \mathbf{c}_t = \sum_{s \in \mathcal{S}} b_s \, \mathbf{e}_s, \quad (14)$$

where $\{a_s\}$ and $\{b_s\}$ are expected and realized discounted visit-counts.

Given only these state-indexed accumulation vectors, overlap should arise exclusively from matching state indices. Without additional structural information such as reachability, it is unreasonable to attribute overlap between distinct basis elements $(\mathbf{e}_{s_i}, \mathbf{e}_{s_j})$, $i \neq j$. Therefore, the inner product $\langle \mathbf{m}(s_t, a_t), \mathbf{c}_t \rangle = \sum_{s \in \mathcal{S}} a_s b_s$ naturally quantifies the overlap between future and past visitation. To obtain a scale-stable redundancy measure suitable for exploration, we further normalize this measure as follows.

**Definition 3.2** (Tabular FPVR). Let $\tau = (s_0, s_1, \dots)$ be a trajectory generated by a policy interacting with an MDP $\mathcal{M} = (\mathcal{S}, \mathcal{A}, P, r, \gamma)$. At current time step $t$, define the SR vector $\mathbf{m}(s_t, a) \in \mathbb{R}^{|\mathcal{S}|}$ with components $\mathbf{m}(s_t, a)[s'] = M(s_t, a, s')$, and the PR vector $\mathbf{c}_t \in \mathbb{R}^{|\mathcal{S}|}$ with components $\mathbf{c}_t[s'] = C_t(s')$ for $a \in \mathcal{A}, s' \in \mathcal{S}$. The *future–past visitation redundancy* (FPVR) of the state-action pair $(s_t, a)$ in the tabular setting is defined as the cosine similarity

$$\mathcal{F}^{(t)}(s_t, a) \doteq \frac{\langle \mathbf{m}(s_t, a), \mathbf{c}_t \rangle}{\|\mathbf{m}(s_t, a)\|_2 \, \|\mathbf{c}_t\|_2}. \quad (15)$$

In this paper, at time $t$, we only focus on $\mathcal{F}^{(t)}$ of the current state $s_t$ to select $a_t$. FPVR values across actions at a fixed state can be very close, making the FPVR bias too weak to meaningfully affect action selection. We therefore apply an action-wise z-score normalization to amplify the relative FPVR differences across actions:

$$\bar{\mathcal{F}}^{(t)}(s_t, a) = \frac{\mathcal{F}^{(t)}(s_t, a) - \mu_{\mathcal{F}}(s_t)}{\sigma_{\mathcal{F}}(s_t) + \varepsilon}, \quad (16)$$

where $\mu_{\mathcal{F}}(s_t)$ and $\sigma_{\mathcal{F}}(s_t)$ denote the mean and standard deviation of $\{\mathcal{F}^\pi(s_t, a')\}_{a' \in \mathcal{A}}$ over the action space, and $\varepsilon > 0$ is a small constant for numerical stability.

To use (16) as an exploration signal, we incorporate it into action selection via a biased $Q$ function

$$Q_b^{(t)}(s_t, a) = Q(s_t, a) - \alpha \, \bar{\mathcal{F}}^{(t)}(s_t, a) \quad (17)$$

where $\alpha > 0$ is a bias coefficient. Actions are selected using a standard action-selection rule, such as $\epsilon$-greedy or Boltzmann exploration, applied to $Q_b^{(t)}(s, a)$. In this way, FPVR

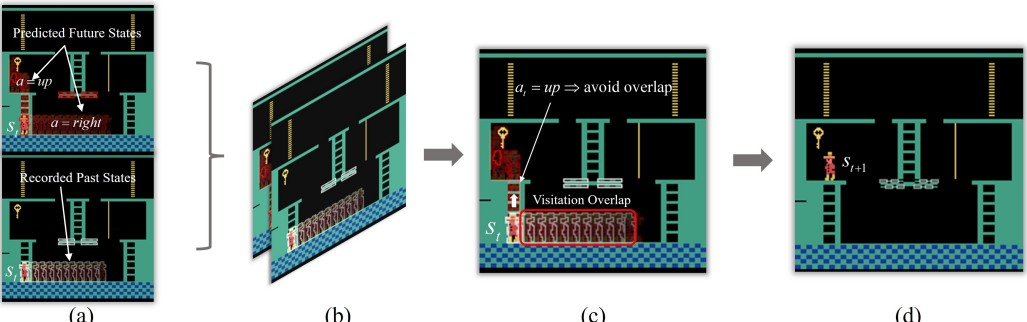

(a)          (b)          (c)          (d)

*Figure 1.* Illustration of FPVR-based exploration in the Atari game *Montezuma's Revenge*. (a) At the current state $s_t$, the agent constructs a representation for the $(s_t, a)$-conditioned future discounted accumulative state features and another representation recording the recent past discounted accumulative state features. The shade of the contour indicates the extent to which the corresponding state features are discounted. (b) These two representations constitute the dual-timescale state representation. (c) The future-past overlap of each action (e.g. *up* and *right*) is measured through FPVR. (d) The agent executes the action *up* that results in the least overlap and transitions to $s_{t+1}$.

discourages actions whose predicted future visitation overlaps strongly with recently visited states, while remaining fully compatible with conventional value-based decision. In Appendix A.2, Proposition A.2 further gives an idealized analysis showing that uniformly controlling the (unnormalized) future–past overlap can imply an $O(n)$ upper bound on the expected cover time.

**Proposition 3.3.** *Consider a scenario where the immediate reward depends only on the state visited at the current time step $t$, i.e., $r_t = r(s_t)$. Define the unnormalized tabular FPVR*

$$Q_{\mathcal{F}}^{(t)}(s_t, a) \doteq \langle \mathbf{m}(s_t, a), \mathbf{c}_t \rangle = \sum_{s' \in \mathcal{S}} M(s_t, a, s')\, C_t(s').$$

$$(18)$$

*Then $Q_{\mathcal{F}}^{(t)}(s_t, a)$ is exactly the action-value function induced by a history-dependent intrinsic reward*

$$r_{\mathrm{FP}}^{(t)}(s) \doteq C_t(s). \qquad (19)$$

*Proof.* This proposition is an immediate consequence based on the SR decomposition of the $Q$ function in (6). $\square$

**Difference between FPVR exploration and its equivalent intrinsic reward exploration.** The equivalent intrinsic reward (18) naturally suggests an intrinsic-reward variant of FPVR that uses $r_{\mathrm{FP}}^{(t)}(s)$ to guide exploration. However, compared with this intrinsic-reward formulation, our FPVR scheme has two advantages: (i) Since $r_{\mathrm{FP}}^{(t)}(s)$ is history-dependent, intrinsic-reward methods must repeatedly re-propagate its changes through value iteration, whereas FPVR propagates the reward immediately via SR, resulting in more timely updates of the exploration policy (demonstrated in Section 4.1). (ii) The learned SR can still be used for exploration even if it has stopped update, and only updating the PR can guide the agent to explore with high coverage efficiency (demonstrated in Appendix D.1).

### 3.3. FPVR with Function Approximation

We now extend FPVR from the tabular setting to high-dimensional or continuous state spaces using function approximation. The key challenge is that, unlike one-hot state encodings, general feature representations are not orthogonal for different states, which fundamentally alters how the future-past overlap should be measured.

Let $\phi : \mathcal{S} \to \mathbb{R}^d$ denote a feature map. The SR is generalized to *successor features* (SF) as shown in (7), which encode the expected discounted accumulation of future state features. Correspondingly, we introduce *persistence features* as a general version of PR.

**Definition 3.4** (Persistence Features). Let $\tau = (s_0, s_1, \dots)$ be a state trajectory generated by a policy interacting with the MDP $\mathcal{M} = (\mathcal{S}, \mathcal{A}, P, r, \gamma)$. For $\lambda \in (0, 1)$, the *persistence features* (PF) are defined as the discounted accumulation of past state features:

$$\chi_t \doteq \sum_{k=0}^{t} \lambda^{t-k} \phi(s_k) \in \mathbb{R}^d. \qquad (20)$$

Equivalently, with $\chi_{-1} \doteq \mathbf{0}$, PF admits the recursion

$$\chi_t = \phi(s_t) + \lambda\, \chi_{t-1}. \qquad (21)$$

When $\phi(s)$ is the one-hot encoding of state $s$, PF reduces exactly to the tabular PR defined in Definition 3.1.

A direct extension of (15) would compute cosine similarity between the SF $\psi(s_t, a)$ and the PF $\chi_t$, but this is inappropriate in general feature spaces. Unlike one-hot encodings, learned features often contain shared dimensions (e.g., background or low-level visual components) that activate across many states (Omama et al., 2025). Thus $\langle \psi(s_t, a), \chi_t \rangle$ includes not only true overlap from the same underlying states but also spurious cross-terms from co-activated, unrelated states. Consequently, cosine similarity can overestimate the true overlap between future and past state visitation.

**ZCA whitening via online statistics.** Assume that encoder features $\phi(s)$ follow a fixed reference distribution $\nu$ with mean $\mu_\nu$ and covariance $\Sigma_\nu$. The (regularized) ZCA transform

$$\tilde{\phi}(s) = (\Sigma_\nu + \varepsilon I)^{-1/2} (\phi(s) - \mu_\nu) \tag{22}$$

satisfies $\mathbb{E}_\nu[\tilde{\phi}(s)] = \mathbf{0}$ and $\mathrm{Cov}_\nu(\tilde{\phi}(s)) \approx I$, so that $\mathbb{E}_\nu\left[\|\tilde{\phi}(s)\|_2^2\right] = \mathrm{tr}(I) = d$ and inner products become scale- and correlation-controlled similarity scores rather than being dominated by a few high-variance, co-activated dimensions (Kessy et al., 2018; Huang et al., 2018). In FPVR, however, the compared quantities are weighted sums of *unwhitened* features: $\chi_t = \sum_{k=0}^{t} \lambda^{t-k}\phi(s_k)$ and $\psi(s_t, a) = \mathbb{E}\big[\sum_{k=0}^{\infty}\gamma^k\phi(S_k)\,\big|\,S_0 = s_t, A_0 = a\big]$. The constituent $\phi(s_k)$ are drawn from a sequence of policy- and horizon-dependent visitation distributions, so a single global reference $(\mu_\nu, \Sigma_\nu)$ that is simultaneously ideal for all $t$ is generally unattainable. Therefore, at each time step $t$, we compute whitening statistics $(\mu_t, \Sigma_t)$ w.r.t. a reference distribution $\nu_t$ associated with the data stream (e.g., an empirical distribution of recently observed features stored in a buffer), and treat them as online estimates rather than the exact moments of any stationary $\pi$-induced distribution. We then define the time-varying ZCA-whitened feature as

$$\tilde{\phi}^{(t)}(s) = (\Sigma_t + \varepsilon I)^{-1/2} (\phi(s) - \mu_t), \tag{23}$$

where $\varepsilon > 0$ is a small constant for numerical stability.

Accordingly, we define the whitened SF at time $t$ as

$$\tilde{\psi}^{(t)}(s, a) = \mathbb{E}\left[\sum_{k=0}^{\infty}\gamma^k\,\tilde{\phi}^{(t)}(S_k)\,\middle|\,S_0 = s, A_0 = a\right], \tag{24}$$

and the whitened PF as

$$\tilde{\chi}_t^{(t)} = \sum_{k=0}^{t}\lambda^{t-k}\,\tilde{\phi}^{(t)}(s_k). \tag{25}$$

**Definition 3.5** (FPVR with Function Approximation). Let $\tau = (s_0, s_1, \dots)$ be a trajectory generated by a policy interacting with an MDP $\mathcal{M} = (\mathcal{S}, \mathcal{A}, P, r, \gamma)$. At current time step $t$, the *future–past visitation redundancy* (FPVR) of the state-action pair $(s_t, a)$ under function approximation is defined as the cosine similarity between the whitened SF (24) and the whitened PF (25)

$$\tilde{\mathcal{F}}^{(t)}(s_t, a_t) \doteq \frac{\tilde{\psi}^{(t)}(s_t, a)^\top \tilde{\chi}_t^{(t)}}{\|\tilde{\psi}^{(t)}(s_t, a)\|_2 \, \|\tilde{\chi}_t^{(t)}\|_2}. \tag{26}$$

Figure 2 illustrates the general procedure for computing FPVR in the function approximation setting. For practical implementation, the z-score (16) is also applied to $\tilde{\mathcal{F}}^{(t)}(s_t, a)$ to obtain a biased $Q$ function as in (17).

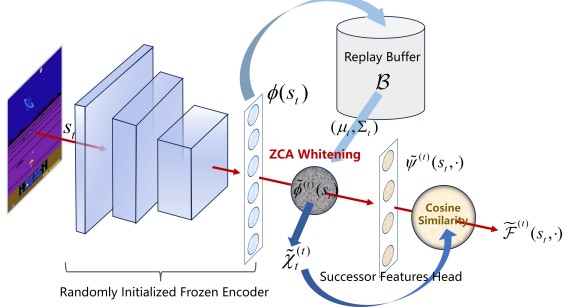

*Figure 2.* General FPVR pipeline. A randomly initialized frozen encoder maps the observation $s_t$ to feature $\phi(s_t)$, whose running mean and covariance $(\mu_t, \Sigma_t)$ are estimated from the replay buffer $\mathcal{B}$ to perform ZCA whitening and obtain $\tilde{\phi}^{(t)}(s_t)$. The whitened feature is accumulated into PF $\tilde{\chi}_t^{(t)}$, while a SF head predicts $\tilde{\psi}^{(t)}(s_t, \cdot)$. The FPVR score $\tilde{\mathcal{F}}_t^{(t)}(s_t, \cdot)$ is computed as the cosine similarity between $\tilde{\psi}(s_t, \cdot)$ and $\tilde{\chi}_t$.

Similar to the tabular case, the unnormalized FPVR $\tilde{\psi}^{(t)}(s_t, a)^\top \tilde{\chi}_t^{(t)}$ can also be interpreted as a $Q$ function according to (8) with the equivalent intrinsic reward

$$r_{\mathrm{FP}}^{(t)}(s) = \tilde{\phi}^{(t)}(s)^\top \tilde{\chi}_t^{(t)} = \sum_{k=0}^{t}\lambda^{t-k}\,\tilde{\phi}^{(t)}(s)^\top \tilde{\phi}^{(t)}(s_k), \tag{27}$$

which reduces to $C_t(s)$ in the tabular setting. This equivalent reward encourages the agent to visit states that have low inner product similarity with recent trajectories, and feature whitening is a prerequisite for enabling the inner product to effectively measure similarity (Omama et al., 2025).

**Connection to pseudo-count-based intrinsic reward.** Bellemare et al. (2016) showed that a learned density model can induce state-visit pseudo-counts, which in turn yield intrinsic rewards for exploration. Proposition A.1 in Appendix A.1 further shows that, under mild boundedness and decorrelated-aggregation assumptions, the pseudo-count $\hat{N}_t(s)$ induced by the shifted FPVR-based density model is asymptotically a strictly increasing rational function of the equivalent reward $r_{\mathrm{FP}}^{(t)}(s)$. Thus any classical pseudo-count bonus that is decreasing in $\hat{N}_t(s)$ (e.g., $\hat{N}_t(s)^{-1/2}$) can be seen as a strictly decreasing function of $r_{\mathrm{FP}}^{(t)}(s)$, i.e., an increasing function of the negative overlap $-r_{\mathrm{FP}}^{(t)}(s)$, formally linking FPVR's suppression of future–past visitation overlap to count-based notions of novelty.

### 3.4. Practical Implementation with DQN

Since FPVR only adds a bias on action values ((17)) for exploration, it can be integrated into any RL method that performs policy improvement via a learned $Q$ function, including value-based algorithms such as DQN and actor–critic methods (e.g., DDPG and SAC) (Mnih et al., 2015; Lillicrap et al., 2015; Haarnoja et al., 2018). For a fair com-

parison with SR-based exploration baselines in *Atari 2600* games (Machado et al., 2018a; 2020; Yu et al., 2023), we implement DQN+FPVR by following the DQN+MMC pipeline of Machado et al. (2020) and using the mixed Monte-Carlo (MMC) return (Bellemare et al., 2016; Ostrovski et al., 2017). As shown in Figure 2, the introduction of FPVR adds only a dual-timescale module consisting of a whitened SF predictor and a whitened PF accumulator.

**Whitened SF and PF.** Every $K$ environment steps, we estimate the whitening statistics from a buffer of recent encoder features $\mathcal{B}_t$ by

$$\hat{\mu}_t = \frac{1}{|\mathcal{B}_t|} \sum_{s \in \mathcal{B}_t} \phi(s), \hat{\Sigma}_t = \frac{1}{|\mathcal{B}_t|} \sum_{s \in \mathcal{B}_t} \big(\phi(s) - \hat{\mu}_t\big)\big(\phi(s) - \hat{\mu}_t\big)^\top,$$
(28)

then update them by EMA

$$\mu_t = (1-\rho)\,\mu_{t-1} + \rho\,\hat{\mu}_t, \qquad \Sigma_t = (1-\rho)\,\Sigma_{t-1} + \rho\,\hat{\Sigma}_t,$$
(29)

where $\rho \in (0,1)$ is the EMA learning rate, yielding the ZCA-whitened feature

$$\tilde{\phi}^{(t)}(s) = (\Sigma_t + \varepsilon I)^{-1/2}\big(\phi(s) - \mu_t\big).$$
(30)

The whitened PF is updated with

$$\tilde{\chi}_t^{(t)} = \begin{cases} \mathbf{0}, & \text{if } t \text{ is the start of an episode,} \\ \lambda\,\tilde{\chi}_{t-1}^{(t-1)} + \tilde{\phi}^{(t)}(s_t), & \text{otherwise.} \end{cases}$$
(31)

At time step $t$, we train a lightweight whitened-SF head $\tilde{\psi}_\theta^{(t)}(s,a)$ from the replay buffer $\mathcal{D}$ (without a separate SF target network) using the TD target

$$y^{\tilde{\psi}} = \tilde{\phi}^{(t)}(s) + \gamma(1-d)\,\tilde{\psi}_\theta^{(t)}(s', \bar{a}),$$
(32)

where $\bar{a} = \arg\min_{a \in \mathcal{A}} \tilde{\mathcal{F}}^{(t)}(s', a)$, by minimizing

$$\mathcal{L}_{\text{SF}}(\theta) = \mathbb{E}_{(s,a,s',d) \sim \mathcal{D}}\left[\Big\|\tilde{\psi}_\theta^{(t)}(s,a) - y^{\tilde{\psi}}\Big\|_2^2\right].$$
(33)

**Exploration via a biased $Q$ function.** At time $t$,

$$\tilde{\mathcal{F}}_t^{(t)}(s_t, a) = \frac{\big(\tilde{\psi}_\theta^{(t)}(s_t, a)\big)^\top \tilde{\chi}_t^{(t)}}{\|\tilde{\psi}_\theta^{(t)}(s_t, a)\|_2 \,\|\tilde{\chi}_t^{(t)}\|_2}.$$

We apply the z-score (16) to obtain $\tilde{\bar{\mathcal{F}}}^{(t)}(s_t, a)$ and then form the biased action values via (17), denoted by $Q_b^{(t)}(s_t, a)$. Action selection uses $\epsilon$-greedy on $Q_b^{(t)}(s_t, a)$. The DQN update (including MMC) follows Machado et al. (2020) exactly. Detailed algorithm implementation of DQN+FPVR is provided in Appendix B.

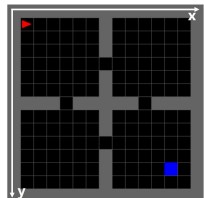 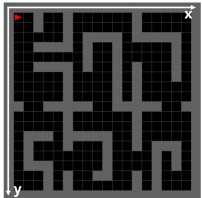

(a) Tabular MiniGrid-FourRooms    (b) Visual MiniGrid-Maze

*Figure 3.* MiniGrid environments. (a) A $15 \times 15$ tabular *FourRooms* environment whose observation is values encoding the agent's position. (b) A $20 \times 20$ visual *Maze* environment whose observation is a raw RGB image.

## 4. Experiments

In this section, we will validate the two claimed advantages of FPVR exploration: high coverage efficiency and the ability to discover novel states. To evaluate coverage efficiency, we consider both the tabular and visual variants of *MiniGrid* (Chevalier-Boisvert et al., 2023). To assess the ability to discover novel states, we consider six exploration-challenging *Atari 2600* games from the Arcade Learning Environment (Bellemare et al., 2013). The baselines considered are primarily successor-representation-based intrinsic exploration methods closely related to FPVR, including Count-Based Exploration with the Successor Representation (SR) (Machado et al., 2020) and Successor-Predecessor Intrinsic Exploration (SP) (Yu et al., 2023), as well as the classical Random Network Distillation (RND) (Burda et al., 2019). Implementation details for the environments, hyperparameters, and network architectures are provided in Appendix C. Unless stated otherwise, all curves report the mean over independent random seeds, and the shaded region indicates $\pm$ one standard deviation.

### 4.1. Evaluation of coverage efficiency

**Tabular FourRooms (exploration-only).** We evaluate coverage in *FourRooms* (Figure 3(a)) by comparing FPVR to SR+SARSA, SP+SARSA, a random walk, and the FPVR-equivalent intrinsic-reward baseline ((19), i.e. $r^{\text{FP}}$+SARSA shown in the figures) implemented with $-C_t(s)$ as the intrinsic reward obtained by visiting $s$. FPVR directly selects actions from the action-wise z-scored FPVR values in (16) via a Boltzmann policy

$$\pi_\beta(a \mid s_t) = \frac{\exp\big(-\beta\big(\bar{\mathcal{F}}^{(t)}(s_t, a) - \min_{a' \in \mathcal{A}} \bar{\mathcal{F}}^{(t)}(s_t, a')\big)\big)}{\sum_{a'' \in \mathcal{A}} \exp\big(-\beta\big(\bar{\mathcal{F}}^{(t)}(s_t, a'') - \min_{a' \in \mathcal{A}} \bar{\mathcal{F}}^{(t)}(s_t, a')\big)\big)}.$$
(34)

We remove the goal and run each method for 3,000 interaction steps. Figure 4 reports results averaged over 50 random seeds. FPVR attains the fastest and most uniform coverage and the smallest cover time; see Proposition A.2 in Appendix A.2 for an idealized theoretical explanation. It also outperforms $r^{FP}$ + SARSA, consistent with SF/SR enabling faster propagation of the history-dependent sig-

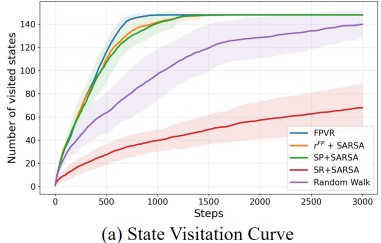 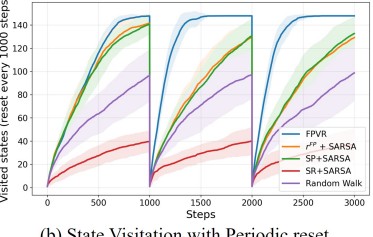 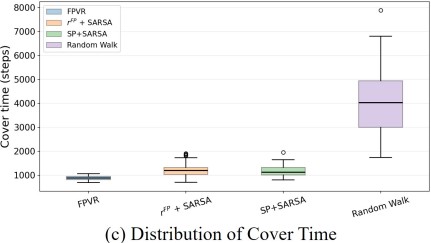

|  (a) State Visitation Curve | (b) State Visitation with Periodic reset | (c) Distribution of Cover Time |

*Figure 4.* Coverage efficiency validation in tabular *FourRooms* without goals. The shaded regions around the curves represent ± std. (a) State visitation curves for 5 different exploration methods. (b) State visitation curves with the visitation counter being reset every 1,000 steps. (c) Cover time distribution with different exploration methods. In each boxplot, the central line denotes the median cover time across trials; the lower and upper edges of the box denote the first and third quartiles (the 25th and 75th percentiles), and the box height is the interquartile range. The whiskers extend to the most extreme values that are not considered outliers under the standard 1.5×IQR convention, while any points beyond the whiskers are plotted as outliers.

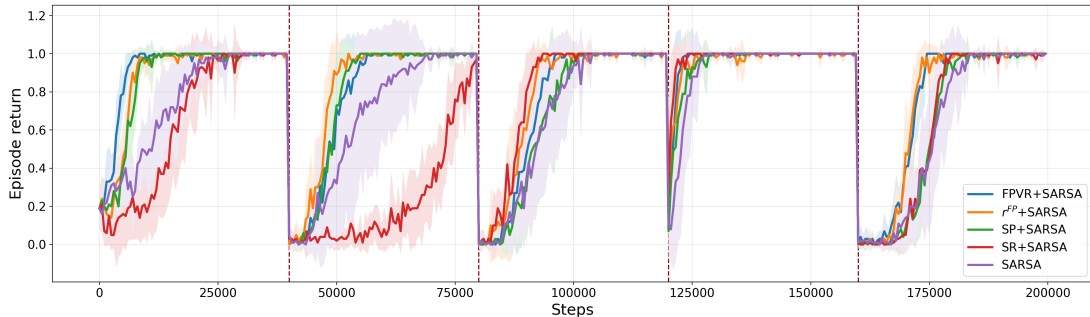

*Figure 5.* Return curves for goal-reaching tasks in tabular *FourRooms*. The initial position of the agent is randomly reset each episode with a maximum episode length of 200. Whenever the goal is reached, the agent obtains a +1 reward and the episode is terminated. The goal is randomly reset every 40,000 steps (while keeping the reset schedule identical across all methods).

nal. Interestingly, the periodically reset visitation curves indicate that FPVR exploration exhibits increasingly higher coverage efficiency as exploration proceeds; we attribute this to SR gradually converging to a pattern that reflects the average dynamics, which is validated by FPVR exploration experiments with a well-learned SR in Appendix D.1.

**FourRooms with non-stationary goals.** Under periodically changing goals, we evaluate adaptation in the goal-reaching task; detailed environment setup and results are shown in Figure 5. Overall, FPVR and $r^{FP}$+SARSA adapt faster than the other baselines after each goal change. This highlights the benefit of rapid re-coverage for learning under non-stationary reward.

**Visual Maze (function approximation).** In the visual *Maze* (Figure 3(b)), FPVR uses (34) while SR and SP are combined with DQN; we also report tabular FPVR on a topology-matched tabular Maze. Figure 6 shows that FPVR achieves consistently higher coverage than the baselines within 9,000 steps. Similar to the tabular case, FPVR exhibits increasingly higher coverage efficiency as the SF network training proceeds; in the visual *Maze* environment, observations are RGB renderings converted to grayscale and resized to $128 \times 128$, and key hyperparameters (e.g., replay buffer size/batch size and the $\epsilon$-schedule for DQN baselines)

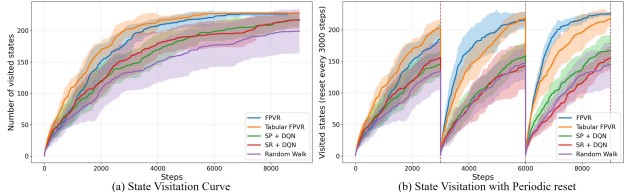

|  (a) State Visitation Curve | (b) State Visitation with Periodic reset |

*Figure 6.* Coverage efficiency evaluation in visual *Maze*. The shaded regions around the curves represent ± std. (a) State visitation curves for 5 different exploration methods. (b) State visitation curves with the visitation counter being reset every 3,000 steps.

follow Appendix C.

### 4.2. Evaluation of ability to discover novel states

To evaluate FPVR's ability to discover novel states, we consider the six sparse-reward, exploration-challenging *Atari 2600* games mentioned in Table 1. The benchmarks established by Machado et al. (2020) and Machado et al. (2018a) provide a convenient and widely used protocol for comparison and evaluation. To ensure a fair comparison, we follow exactly the same environmental setup as Machado et al. (2020), referring to the environment configuration in their open-source code. The mixed Monte-Carlo (MMC) return is also adopted (Bellemare et al., 2016; Ostrovski

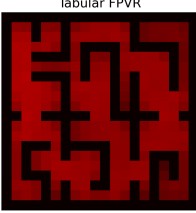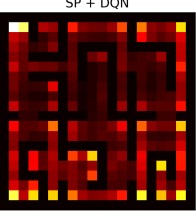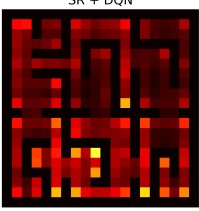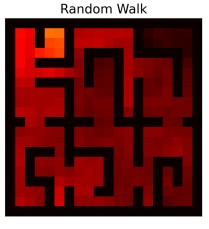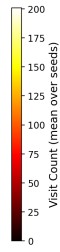

*Figure 7.* Mean position visit count over 10 runs in the visual *Maze* environment.

*Table 1.* Performance comparison on 6 hard-to-explore *Atari 2600* games (values in parentheses denote the standard deviation ).

| Method | Freeway | Gravitar | MontezumaRevenge | PrivateEye | Solaris | Venture |
|---|---|---|---|---|---|---|
| DQN | **32.4 (0.3)** | 118.5 (22.0) | 0.0 (0.0) | **1447.4 (2567.9)** | 783.4 (55.3) | 4.4 (5.4) |
| DQN$^{\text{MMC}}$ | 29.5 (0.1) | 1078.3 (254.1) | 0.0 (0.0) | 113.4 (42.3) | 2244.6 (378.8) | 1220.1 (51.0) |
| RND | 28.2 (0.2) | 790.0 (122.9) | 524.8 (314.0) | 61.3 (53.7) | 1270.3 (291.0) | 953.7 (167.3) |
| SR + DQN$^{\text{MMC}}$ | 29.4 (0.1) | 457.4 (120.3) | 1395.4 (1121.8) | 104.4 (50.4) | 1890.1 (163.1) | 1348.5 (56.5) |
| SP + DQN$^{\text{MMC}}$ | 27.5 (0.2) | 1223.0 (408.9) | 1530.0 (1072.1) | 488.2 (390.9) | 2455.8 (262.0) | 1274.0 (133.2) |
| FPVR + DQN$^{\text{MMC}}$ | 30.9 (0.1) | **1328.8** (119.6) | **1660.0 (1028.8)** | 88.7 (34.0) | **2596.7 (301.3)** | **1384.3 (119.8)** |

et al., 2017). We evaluate the FPVR+DQN$^{\text{MMC}}$ agent proposed in Section 3.4. The detailed hyperparameters and environment settings are provided in Appendix C.3.

The results in Table 1 report the average performance of FPVR+DQN$^{\text{MMC}}$ over 10 seeds at 100M frames under exactly the same environment setup across all methods. The scores of DQN, DQN$^{\text{MMC}}$, RND (except for *Freeway*), and SR+DQN$^{\text{MMC}}$ are taken from Machado et al. (2020), while the scores of SP+DQN$^{\text{MMC}}$ and RND on *Freeway* are taken from Yu et al. (2023). For each seed, the learned policy is evaluated in 30 independent trials, and the average return is used as the final score for that seed. The results show that FPVR+DQN$^{\text{MMC}}$ outperforms the $DQN^{MMC}$ in 5 of the 6 tasks and achieves the best performance among all methods on *MontezumaRevenge*, *Solaris*, *Gravitar* and *Venture*.

## 5. Related Work

Prior work often encourages exploration via intrinsic signals based on novelty or prediction error, including curiosity and prediction residuals (Pathak et al., 2017; Burda et al., 2019), episodic memory/reachability (Savinov et al., 2019), and related directed or stabilized exploration objectives (Badia et al., 2020; Henaff et al., 2022; Castanyer et al., 2024). Complementary lines emphasize rapid traversal and coverage, e.g., cover-time/option-based methods and spectral or graph-based skills (Jinnai et al., 2019; 2020; Klissarov & Machado, 2023), as well as coverage-driven pretraining and scalable covering options (Campos et al., 2021; Chen et al., 2022), at the cost of additional policy machinery (Hutsebaut-Buysse et al., 2022; Bagaria et al., 2023). FPVR instead adds a lightweight bias to the $Q$ function to suppress short-term revisits, combining the strengths of both novelty-driven

and coverage-oriented exploration.

Successor representation (SR) is a predictive-map state representation with links to neuroscience (Dayan, 1993; Stachenfeld et al., 2017) and has inspired SR-based intrinsic exploration (Machado et al., 2020; Yu et al., 2023; Machado et al., 2018b). Our dual-timescale design pairs SR with a persistence representation (PR) conceptually related to visual persistence/iconic memory (Van Kerkoerle et al., 2017; Teeuwen et al., 2021). Further exploring the neuroscientific basis of the dual-timescale state representation (DTSR) and their role in decision making is an interesting direction.

## 6. Conclusion

We proposed *Future–Past Visitation Redundancy* (FPVR), an exploration method that improves state-space coverage efficiency by suppressing short-term revisitation. FPVR is induced by a *Dual-Timescale State Representation* (DTSR) that combines a slowly learned successor representation (SR) / successor features (SF) capturing expected discounted *future* visitation with a rapidly updated persistence representation / persistence features (PR/PF) summarizing discounted *past* visitation. We compute FPVR as a normalized overlap between the two representations, and inject it as a bias on action values to guide exploration without optimizing an additional intrinsic-reward objective. For function approximation, we apply online ZCA whitening to make the overlap metric scale- and correlation-normalized.

Empirically, experiments in *MiniGrid* (tabular and visual) and *Atari 2600* demonstrate that FPVR reduces cover time and improves learning efficiency in sparse-reward and non-stationary settings, while keeping computational overhead comparable to standard value-based baselines.

## Impact Statement

This paper presents work whose goal is to advance the field of Machine Learning. There are many potential societal consequences of our work, none which we feel must be specifically highlighted here.

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

# Appendix Contents

## A. Main Theoretical Results

### A.1. Pseudo-Count Interpretation of the FPVR-Equivalent Intrinsic Reward

**Problem statement and notation.** We study the relationship between the FPVR-equivalent intrinsic signal defined in (27) and the pseudo-count induced by a density model in the sense of count-based exploration.

Following the pseudo-count framework of Bellemare et al. (2016), a density model is a sequence of probability distributions $\{\rho_t(\cdot)\}_{t \geq 0}$ over the state space $\mathcal{S}$, where $\rho_t(s)$ denotes the model's probability of state $s$ after observing a trajectory of length $t$. If the model is *learning-positive* at $s$ in the sense that the updated probability $\rho'_t(s)$ after appending one more occurrence of $s$ satisfies

$$0 < \rho_t(s) < \rho'_t(s) < 1,$$

then there exist nonnegative pseudo-counts $\hat{N}_t(s)$ and a pseudo total count $\hat{n}_t = \sum_x \hat{N}_t(x)$ such that

$$\rho_t(s) = \frac{\hat{N}_t(s)}{\hat{n}_t}, \qquad \rho'_t(s) = \frac{\hat{N}_t(s) + 1}{\hat{n}_t + 1}. \tag{35}$$

Solving this system yields the closed-form expression

$$\hat{N}_t(s) = \frac{\rho_t(s)\big(1 - \rho'_t(s)\big)}{\rho'_t(s) - \rho_t(s)}. \tag{36}$$

Our goal is to relate $\hat{N}_t(s)$ to the FPVR-equivalent intrinsic signal $q^{(t)}(s)$ defined below, which coincides (up to dropping the action argument) with (27).

We now introduce the FPVR notation used in this subsection. For each time $t$, let $\tilde{\phi}^{(t)} : \mathcal{S} \to \mathbb{R}^d$ be the time-varying whitened feature map constructed from the online statistics $(\mu_t, \Sigma_t)$ as defined in Sec. 3.3. In the full FPVR algorithm, the inner product signal involves the whitened successor features $\tilde{\psi}^{(t)}(s, a)$ and the persistent feature $\tilde{\chi}_t^{(t)}$.

Fix a discount factor $\lambda \in [0, 1)$. For a trajectory $\tau = (s_0, \ldots, s_t)$, define the discounted persistence feature

$$\tilde{\chi}_t^{(t)} := \sum_{k=0}^{t} \lambda^{t-k} \, \tilde{\phi}^{(t)}(s_k). \tag{37}$$

For any $s \in \mathcal{S}$, define the discounted overlap

$$q^{(t)}(s) := \big(\tilde{\phi}^{(t)}(s)\big)^\top \tilde{\chi}_t^{(t)} = \sum_{k=0}^{t} \lambda^{t-k} \big(\tilde{\phi}^{(t)}(s)\big)^\top \tilde{\phi}^{(t)}(s_k). \tag{38}$$

By (27), $q^{(t)}(s)$ coincides with the FPVR-equivalent intrinsic signal.

To connect FPVR with a density model, we introduce an auxiliary, constructed density on a finite, time-varying support set. For each $t$, let $\mathcal{S}_t \subset \mathcal{S}$ be a finite set of states (for example, the set of states currently present in the replay buffer at time $t$), and assume that all sums below are taken over $\mathcal{S}_t$. We further assume that the whitened features are uniformly bounded: there exists $L > 0$ such that

$$\big\|\tilde{\phi}^{(t)}(x)\big\|_2 \leq L \quad \text{for all } x \in \mathcal{S}, \text{ all } t. \tag{39}$$

Then, by (37),

$$\big\|\tilde{\chi}_t^{(t)}\big\|_2 \leq \sum_{k=0}^{t} \lambda^{t-k} \big\|\tilde{\phi}^{(t)}(s_k)\big\|_2 \leq L \sum_{j=0}^{\infty} \lambda^j = \frac{L}{1 - \lambda}, \tag{40}$$

and by Cauchy–Schwarz,

$$\big|q^{(t)}(s)\big| = \big|\big(\tilde{\phi}^{(t)}(s)\big)^\top \tilde{\chi}_t^{(t)}\big| \leq \big\|\tilde{\phi}^{(t)}(s)\big\|_2 \big\|\tilde{\chi}_t^{(t)}\big\|_2 \leq \frac{L^2}{1 - \lambda} \quad \text{for all } s, t. \tag{41}$$

We introduce a global positive shift. Let $K := L^2/(1 - \lambda)$ and choose any constant

$$B > K. \tag{42}$$

Define the shifted scores

$$\tilde{q}^{(t)}(s) := q^{(t)}(s) + B. \tag{43}$$

By (41) and (42), we have $\tilde{q}^{(t)}(s) \geq B - K > 0$ for all $s, t$.

We now define a density model on the finite support $\mathcal{S}_t$ via

$$Z_t := \sum_{x \in \mathcal{S}_t} \tilde{q}^{(t)}(x) = \sum_{x \in \mathcal{S}_t} \left( q^{(t)}(x) + B \right), \tag{44}$$

and

$$\rho_t(s) := \frac{\tilde{q}^{(t)}(s)}{Z_t} = \frac{q^{(t)}(s) + B}{Z_t}, \qquad s \in \mathcal{S}_t, \tag{45}$$

whenever $Z_t \in (0, \infty)$. By construction, $\rho_t$ is a valid probability mass function on $\mathcal{S}_t$.

To define the recoding probability at $s$, we follow the pseudo-count framework and consider the hypothetical update where one additional copy of $s$ is appended to the data at time $t$ (while keeping the support $\mathcal{S}_t$ fixed). The corresponding persistence feature (with whitening statistics frozen at time $t$) is

$$\tilde{\chi}'_t = \lambda \tilde{\chi}^{(t)}_t + \tilde{\phi}^{(t)}(s), \tag{46}$$

so that

$$q'_t(s) := \left( \tilde{\phi}^{(t)}(s) \right)^\top \tilde{\chi}'_t = \lambda q^{(t)}(s) + c_0(s), \qquad c_0(s) := \left\| \tilde{\phi}^{(t)}(s) \right\|_2^2 > 0, \tag{47}$$

and, for any $x \in \mathcal{S}_t$,

$$q'_t(x) := \left( \tilde{\phi}^{(t)}(x) \right)^\top \tilde{\chi}'_t = \lambda q^{(t)}(x) + \left( \tilde{\phi}^{(t)}(x) \right)^\top \tilde{\phi}^{(t)}(s). \tag{48}$$

Define the interaction term

$$C_t(s) := \sum_{x \in \mathcal{S}_t} \left( \tilde{\phi}^{(t)}(x) \right)^\top \tilde{\phi}^{(t)}(s). \tag{49}$$

Let $n_t := |\mathcal{S}_t|$ and $D_t := B n_t$. Then

$$Z_t = \sum_{x \in \mathcal{S}_t} \left( q^{(t)}(x) + B \right) = \sum_{x \in \mathcal{S}_t} q^{(t)}(x) + D_t. \tag{50}$$

After the hypothetical recoding at $s$, the updated normalizer is

$$\begin{aligned}
Z'_t &:= \sum_{x \in \mathcal{S}_t} \left( q'_t(x) + B \right) \tag{51} \\
&= \sum_{x \in \mathcal{S}_t} \left( \lambda q^{(t)}(x) + (\tilde{\phi}^{(t)}(x))^\top \tilde{\phi}^{(t)}(s) + B \right) \\
&= \lambda \sum_{x \in \mathcal{S}_t} q^{(t)}(x) + C_t(s) + D_t \\
&= \lambda (Z_t - D_t) + C_t(s) + D_t \\
&= \lambda Z_t + C_t(s) + (1 - \lambda) D_t.
\end{aligned}$$

The recoding probability at $s$ is therefore

$$\rho'_t(s) := \frac{q'_t(s) + B}{Z'_t} = \frac{\lambda q^{(t)}(s) + c_0(s) + B}{\lambda Z_t + C_t(s) + (1 - \lambda) D_t}. \tag{52}$$

**Assumptions and rationale.** We now state the assumptions under which the pseudo-count induced by $\rho_t$ admits a clean asymptotic relation to the FPVR signal.

**(A1) Bounded whitened features and positive shift.** The feature bound (39) and choice of $B$ in (42) imply (41) and guarantee that $\tilde{q}^{(t)}(s) = q^{(t)}(s) + B > 0$ for all $s, t$. Consequently, the density model $\rho_t$ in (45) is well-defined for any $t$ such that $0 < Z_t < \infty$.

**(A2) Growing finite support and decorrelated aggregation.** For each $t$, the support $\mathcal{S}_t$ is finite, and its size $n_t = |\mathcal{S}_t|$ grows unboundedly as $t \to \infty$. Let $D_t := Bn_t$. We assume:

*(A2.i) Vanishing average overlap.*

$$\sum_{x \in \mathcal{S}_t} q^{(t)}(x) = o(D_t) \quad \text{as } t \to \infty. \tag{53}$$

*(A2.ii) Vanishing cross-term.*

$$C_t(s) = o(D_t) \quad \text{as } t \to \infty. \tag{54}$$

These conditions capture the intended regime where ZCA-whitened features are approximately zero-mean and weakly correlated across the large support, so cross-terms cancel in aggregate, and the constant shift $B$ dominates the total mass at scale $Z_t \approx D_t$. In particular, since $Z_t = D_t + o(D_t)$ and $D_t \to \infty$, we have $Z_t \to \infty$ and $\rho_t(s) \to 0$ for any fixed $s$ with $s \in \mathcal{S}_t$ eventually.

**(A3) Self-overlap dominance at $s$.** There exist constants $\delta > 0$ and $T_0$ such that for all $t \geq T_0$,

$$c_0(s) - (1 - \lambda)\, q^{(t)}(s) \geq \delta. \tag{55}$$

Equivalently, by (47), the unnormalized score at $s$ strictly increases under the hypothetical recoding for all $t \geq T_0$: $q_t'(s) - q^{(t)}(s) \geq \delta$.

**Proposition.** Under the above assumptions, the pseudo-count induced by $\rho_t$ admits an asymptotically explicit dependence on the FPVR overlap $q^{(t)}(s)$ (equivalently, on the FPVR-equivalent intrinsic signal in (27)). In the asymptotic regime, the leading term is a rational function of $q^{(t)}(s)$, and this function is strictly monotone in $q^{(t)}(s)$.

**Proposition A.1** (FPVR intrinsic signal and pseudo-count via a shifted discounted kernel)**.** *Let $\tilde{\phi}^{(t)}$, $\lambda$, $q^{(t)}$, $\tilde{q}^{(t)}$, $Z_t$, $\rho_t$, $C_t(s)$ and $\rho_t'$ be defined as in (37)–(52), and suppose that (A1)–(A3) hold for a fixed state $s \in \mathcal{S}$. Let $\hat{N}_t(s)$ denote the pseudo-count associated with the density model $\rho_t$ at $s$ in the sense of (36), and let $c_0(s) = \left\| \tilde{\phi}^{(t)}(s) \right\|_2^2$.*

*Define $\eta_t := Z_t'/Z_t$. Then $\eta_t \to 1$ as $t \to \infty$, and*

$$\hat{N}_t(s) = \frac{q^{(t)}(s) + B}{c_0(s) - (1 - \lambda)\, q^{(t)}(s)} \big(1 + o(1)\big) \quad \text{as } t \to \infty. \tag{56}$$

*In particular, the rational map*

$$g(q) := \frac{q + B}{c_0(s) - (1 - \lambda)\, q} \tag{57}$$

*is strictly increasing in $q$ on the domain where the denominator is positive. Consequently, for sufficiently large $t$ (so that the $1 + o(1)$ factor is positive) and any two values $q_1 < q_2$ in this domain,*

$$g(q_1) < g(q_2) \quad \Longrightarrow \quad \hat{N}_t(s; q_1) < \hat{N}_t(s; q_2), \tag{58}$$

*so the pseudo-count $\hat{N}_t(s)$ is asymptotically a strictly increasing function of the FPVR-equivalent intrinsic signal $q^{(t)}(s)$.*

*Proof.* The proof has three parts: we first show that $\rho_t$ is a proper density model and becomes learning-positive at $s$ for large $t$, then obtain asymptotic expressions for $\rho_t(s)$ and $\rho_t'(s)$, and finally use (36) to derive the claimed asymptotic relation and monotonicity.

**Step 1: Proper density model and learning-positivity.** By (A1) and the choice of $B$ in (42), we have $\tilde{q}^{(t)}(x) = q^{(t)}(x) + B > 0$ for all $x, t$. Since $\mathcal{S}_t$ is finite, the normalizer $Z_t$ in (44) is strictly positive and finite, and (45) defines a valid probability mass function on $\mathcal{S}_t$.

By (A2.i), $\sum_{x \in \mathcal{S}_t} q^{(t)}(x) = o(D_t)$, and since $Z_t = \sum_{x \in \mathcal{S}_t} q^{(t)}(x) + D_t$, we obtain

$$Z_t = D_t\big(1 + o(1)\big), \qquad \frac{D_t}{Z_t} \longrightarrow 1 \quad \text{as } t \to \infty. \tag{59}$$

By (A2.ii), $C_t(s) = o(D_t)$, hence also $C_t(s)/Z_t \to 0$. Using (51), we get

$$\eta_t = \frac{Z'_t}{Z_t} = \lambda + \frac{C_t(s)}{Z_t} + (1-\lambda)\frac{D_t}{Z_t} \longrightarrow \lambda + 0 + (1-\lambda) \cdot 1 = 1, \tag{60}$$

hence $\eta_t \to 1$.

Next, by (47) and (A3),

$$q'_t(s) - q^{(t)}(s) = c_0(s) - (1-\lambda)\,q^{(t)}(s) \geq \delta \quad \text{for all } t \geq T_0, \tag{61}$$

and therefore $\tilde{q}'_t(s) > \tilde{q}^{(t)}(s)$ for all $t \geq T_0$. Consider the ratio

$$\frac{\rho'_t(s)}{\rho_t(s)} = \frac{\tilde{q}'_t(s)}{\tilde{q}^{(t)}(s)} \cdot \frac{Z_t}{Z'_t} = \frac{\tilde{q}'_t(s)}{\tilde{q}^{(t)}(s)} \cdot \frac{1}{\eta_t}. \tag{62}$$

The first factor is strictly larger than 1 for all $t \geq T_0$, while the second factor converges to 1 because $\eta_t \to 1$. Hence there exists $T_1 \geq T_0$ such that for all $t \geq T_1$, $\rho'_t(s) > \rho_t(s)$ holds. Moreover, by (59) and boundedness of $q^{(t)}(s)$, (41), we have $\rho_t(s) \to 0$ and $\rho'_t(s) \to 0$, so $0 < \rho_t(s) < \rho'_t(s) < 1$ for all sufficiently large $t$. Thus the model is learning-positive at $s$ in the sense required by (36).

**Step 2: Asymptotic forms of $\rho_t(s)$ and $\rho'_t(s)$.** By definition,

$$\rho_t(s) = \frac{q^{(t)}(s) + B}{Z_t}. \tag{63}$$

Using $\eta_t = Z'_t/Z_t$, the recoding probability can be written as

$$\rho'_t(s) = \frac{\lambda q^{(t)}(s) + c_0(s) + B}{Z'_t} = \frac{\lambda q^{(t)}(s) + c_0(s) + B}{\eta_t Z_t}. \tag{64}$$

By (59) and boundedness of $q^{(t)}(s)$ from (41), we have $\rho_t(s) \to 0$ and $\rho'_t(s) \to 0$, hence $1 - \rho'_t(s) = 1 + o(1)$.

For the key difference, we compute

$$Z_t\big(\rho'_t(s) - \rho_t(s)\big) = \frac{\lambda q^{(t)}(s) + c_0(s) + B}{\eta_t} - \big(q^{(t)}(s) + B\big). \tag{65}$$

Since $\eta_t \to 1$, we have $1/\eta_t = 1 + o(1)$, and the numerator is $O(1)$ by (41). Therefore

$$\begin{aligned} Z_t\big(\rho'_t(s) - \rho_t(s)\big) &= \big(\lambda q^{(t)}(s) + c_0(s) + B\big)\big(1 + o(1)\big) - \big(q^{(t)}(s) + B\big) \\ &= c_0(s) - (1-\lambda)\,q^{(t)}(s) + o(1). \end{aligned} \tag{66}$$

By (A3), the leading term $c_0(s) - (1-\lambda)q^{(t)}(s)$ is bounded below by $\delta > 0$ for all large $t$, so the denominator in (36) is asymptotically well-behaved.

**Step 3: Asymptotic relation and monotonicity in $q^{(t)}(s)$.** Recall (36):

$$\hat{N}_t(s) = \frac{\rho_t(s)\big(1 - \rho'_t(s)\big)}{\rho'_t(s) - \rho_t(s)}. \tag{36'}$$

Since $1 - \rho'_t(s) = 1 + o(1)$, we have

$$\hat{N}_t(s) = \frac{\rho_t(s)}{\rho'_t(s) - \rho_t(s)}\big(1 + o(1)\big). \tag{67}$$

Using (63) and $\rho'_t(s) - \rho_t(s) = \frac{1}{Z_t}Z_t(\rho'_t(s) - \rho_t(s))$, we obtain

$$\hat{N}_t(s) = \frac{Z_t\rho_t(s)}{Z_t(\rho'_t(s) - \rho_t(s))}\big(1 + o(1)\big) = \frac{q^{(t)}(s) + B}{Z_t(\rho'_t(s) - \rho_t(s))}\big(1 + o(1)\big). \tag{68}$$

Substituting (66) yields

$$\hat{N}_t(s) = \frac{q^{(t)}(s) + B}{c_0(s) - (1 - \lambda)\, q^{(t)}(s)}\big(1 + o(1)\big), \tag{69}$$

which is (56).

To see the monotonicity, define $g$ as in (57):

$$g(q) = \frac{q + B}{c_0(s) - (1 - \lambda)\, q}.$$

On the domain where the denominator is positive, $g$ is differentiable and its derivative is

$$g'(q) = \frac{\big(c_0(s) - (1 - \lambda)q\big) - (q + B)\big(-(1 - \lambda)\big)}{\big(c_0(s) - (1 - \lambda)q\big)^2} \tag{70}$$

$$= \frac{c_0(s) + (1 - \lambda)B}{\big(c_0(s) - (1 - \lambda)q\big)^2}.$$

The numerator is strictly positive because $c_0(s) > 0$, $B > 0$ and $0 \le 1 - \lambda < 1$, while the denominator is strictly positive by (A3) (which guarantees $c_0(s) - (1 - \lambda)q^{(t)}(s) \ge \delta$ for all sufficiently large $t$). Therefore $g'(q) > 0$ on the domain of interest, and $g$ is strictly increasing in $q$.

Since (56) states that $\hat{N}_t(s)$ equals $g\big(q^{(t)}(s)\big)$ up to a multiplicative factor $1 + o(1)$ that converges to 1, the leading-order dependence of $\hat{N}_t(s)$ on $q^{(t)}(s)$ is strictly increasing, and the induced ordering coincides asymptotically with that of $g$.

This completes the proof. $\qquad\qquad\square$

**Interpretation.** Proposition A.1 shows that, under mild boundedness and decorrelation assumptions, the pseudo-count induced by the shifted FPVR-based density model is asymptotically a strictly increasing rational function of the FPVR-equivalent intrinsic signal $q^{(t)}(s)$. In particular, standard count-based bonuses that decrease with pseudo-count can be interpreted as increasing functions of the negative FPVR overlap, providing a formal link between FPVR and (pseudo-)count-based novelty.

**A.2. Tabular FPVR implies a cover-time upper bound**

**Problem statement.** From a non-graph-theoretic perspective, it is notoriously difficult to establish a direct, rigorous, and quantitatively useful relationship between a policy's local action-selection preference and the global cover time $T_{\mathrm{cov}}$. The cover time depends jointly on the environment's transition structure, the policy-induced long-horizon closed-loop dynamics, and the evolving visited set $\mathcal{V}_t$; consequently, obtaining general bounds typically requires nontrivial Markov-chain/graph-theoretic tools and delicate coupling arguments. The goal of this subsection is therefore not to provide a tight guarantee that holds for arbitrary MDPs, but rather to derive an interpretable upper bound in an idealized tabular setting under strong simplifying assumptions (e.g., a uniform low-overlap budget and a no-forgetting condition). We view the resulting proposition as a conceptual tool that clarifies how suppressing future–past visitation overlap can promote faster state-space coverage.

**Setup and notation.** Let $\mathcal{M} = (\mathcal{S}, \mathcal{A}, P, r, \gamma)$ be a finite MDP with $|\mathcal{S}| = n < \infty$. In the tabular case, we use one-hot state features $\phi(s) = \mathbf{e}_s \in \mathbb{R}^n$. Let $\tau = (s_0, a_0, s_1, a_1, \dots)$ be the trajectory generated by a policy.

Recall the tabular persistence representation used by FPVR. Fix a past-discount parameter $\lambda \in (0, 1)$ and define

$$\mathbf{c}_t := \sum_{k=0}^{t} \lambda^{t-k} \mathbf{e}_{s_k} \in \mathbb{R}^n, \qquad C_t(s) := \mathbf{e}_s^\top \mathbf{c}_t = \sum_{k=0}^{t} \lambda^{t-k} \mathbb{I}\{s_k = s\}. \tag{71}$$

We also define the tabular successor representation (SR) under a policy $\pi$ in the same convention as (3): for any $(s, a)$,

$$\mathbf{m}(s, a) \doteq \mathbf{m}^\pi(s, a) := \mathbb{E}_\pi\left[\sum_{i=0}^{\infty} \gamma^i \mathbf{e}_{S_{t+i}} \,\middle|\, S_t = s,\ A_t = a\right] \in \mathbb{R}^n. \tag{72}$$

In the tabular case, the *unnormalized* future–past overlap (the numerator of tabular FPVR) is

$$u_t(s,a) \doteq u_t^\pi(s,a) := \big(\mathbf{m}(s,a)\big)^\top \mathbf{c}_t = \sum_{x \in \mathcal{S}} m(s,a,x)\, C_t(x), \tag{73}$$

where $m(s,a,x)$ denotes the $x$-th entry of $\mathbf{m}(s,a)$. Because $m(s,a,x) \geq 0$ and $C_t(x) \geq 0$, we always have $u_t(s,a) \geq 0$.

Let $\mathcal{V}_t := \{s_0, \ldots, s_t\}$ be the set of visited states up to time $t$. Define the cover time

$$T_{\mathrm{cov}} := \inf\{t \geq 0 : \mathcal{V}_t = \mathcal{S}\}. \tag{74}$$

For $k \in \{1, \ldots, n\}$, define the discovery times

$$\tau_k := \inf\{t \geq 0 : |\mathcal{V}_t| \geq k\}, \tag{75}$$

so that $\tau_1 = 0$ and $T_{\mathrm{cov}} = \tau_n$. Let the inter-discovery gaps be

$$\Delta_k := \tau_{k+1} - \tau_k, \qquad k = 1, \ldots, n-1. \tag{76}$$

For each $t \geq 0$, we also define the gap from time $t$ to the next new state as

$$\Delta(t) := \inf\{h \geq 1 : S_{t+h} \notin \mathcal{V}_t\}, \tag{77}$$

with the convention $\Delta(t) = +\infty$ if no such $h$ exists.

Finally, let $\mathcal{H}_t := \sigma(S_0, A_0, \ldots, S_t)$ be the history sigma-field.

**Assumption (uniform low-overlap FPVR regime with no forgetting).** Fix an integer $m \geq 1$. Assume the policy used for exploration, denoted $\pi^{\mathrm{FP}}$, satisfies the following *uniform low-overlap* and *no-forgetting* conditions.

There exist constants $\kappa \geq 0$ and $\eta > 0$ such that along the entire trajectory,

$$u_t(s_t, a_t) = \big(\mathbf{m}(s_t, a_t)\big)^\top \mathbf{c}_t \leq \kappa \qquad \text{for all } t \geq 0, \tag{78}$$

and, furthermore, the tabular persistence does not forget previously visited states: for every time $t < T_{\mathrm{cov}}$ and every state $x \in \mathcal{V}_t$,

$$C_t(x) \geq \eta. \tag{79}$$

Finally, we require the feasibility condition

$$\kappa < m\, \eta\, \gamma^m. \tag{80}$$

Intuitively, (78) formalizes the effect of FPVR: the policy selects actions whose predicted discounted future occupancy has small overlap with the discounted occupancy of the past (the tabular counterpart of the FPVR numerator). The no-forgetting condition (79) idealizes the regime where no visited state is completely forgotten in the persistence representation before the cover time, and (80) ensures that the overlap budget is small enough relative to the horizon $m$ and the effective persistence representation discount mass.

The following result shows that a uniform FPVR overlap budget together with discounted no-forgetting implies a uniform lower bound on the probability of discovering a new state within $m$ steps, which in turn yields an $O(n)$ upper bound on the expected cover time.

**Proposition A.2** (Tabular FPVR overlap control implies a cover-time bound). *Let $\mathcal{M}$ be a finite MDP with $|\mathcal{S}| = n$ and discount $\gamma \in (0,1]$. Let $\pi^{\mathrm{FP}}$ be a (possibly non-stationary, history-dependent) policy generating a trajectory $(s_t, a_t)_{t \geq 0}$. Define $\mathbf{c}_t$, $C_t$, $\mathbf{m}$, and the FPVR overlap $u_t$ as in (71)–(73). Assume (78)–(80) hold for some integer $m \geq 1$.*

*Define*

$$p := 1 - \frac{\kappa}{m\, \eta\, \gamma^m} \in (0, 1]. \tag{81}$$

*Then for every $k \in \{1, \ldots, n-1\}$,*

$$\mathbb{E}[\Delta_k] \leq \frac{m}{p}, \tag{82}$$

*and consequently the expected cover time satisfies*

$$\mathbb{E}[T_{\mathrm{cov}}] = \mathbb{E}[\tau_n] \leq (n-1)\frac{m}{p}. \tag{83}$$

*Proof.* The proof has three steps: (i) we relate the FPVR overlap to the probability of *not* discovering a new state within the next $m$ steps; (ii) we convert this into a geometric tail bound for $\Delta_k$ in blocks of length $m$; (iii) we sum the tail to obtain (82) and then (83).

**Step 1: FPVR overlap upper-bounds the probability of staying in the visited set.** Fix any time $t \geq 0$. Let $\mathcal{V}_t$ be the visited set up to time $t$. By (71), we have $C_t(x) = 0$ if $x \notin \mathcal{V}_t$, and $C_t(x) \geq \eta$ for $x \in \mathcal{V}_t$ by the no-forgetting condition (79). Hence, for any random state $X$ measurable with respect to the future trajectory,

$$\mathbb{I}\{X \in \mathcal{V}_t\} \leq \frac{1}{\eta} C_t(X). \tag{84}$$

Now condition on the history $\mathcal{H}_t$ and consider the next $m$ steps. If no new state is discovered in the next $m$ steps, then $S_{t+1}, \ldots, S_{t+m} \in \mathcal{V}_t$. Therefore,

$$\mathbb{P}(\Delta(t) > m \mid \mathcal{H}_t) = \mathbb{P}(S_{t+1+i} \in \mathcal{V}_t \,\forall i = 0, \ldots, m-1 \mid \mathcal{H}_t)$$

$$\leq \frac{1}{m} \mathbb{E}\left[\sum_{i=0}^{m-1} \mathbb{I}\{S_{t+1+i} \in \mathcal{V}_t\} \,\Big|\, \mathcal{H}_t\right] \tag{85}$$

$$\leq \frac{1}{m\,\eta} \mathbb{E}\left[\sum_{i=0}^{m-1} C_t(S_{t+1+i}) \,\Big|\, \mathcal{H}_t\right], \tag{86}$$

where (85) uses the elementary bound $\mathbb{I}\{\forall i,\ E_i\} \leq \frac{1}{m}\sum_i \mathbb{I}\{E_i\}$ and (86) uses (84).

Next, use $\gamma^i \geq \gamma^{m-1}$ for $i \in \{0, \ldots, m-1\}$ to get

$$\sum_{i=0}^{m-1} C_t(S_{t+1+i}) \leq \frac{1}{\gamma^{m-1}} \sum_{i=0}^{m-1} \gamma^i C_t(S_{t+1+i}) \leq \frac{1}{\gamma^{m-1}} \sum_{i=0}^{\infty} \gamma^i C_t(S_{t+1+i}). \tag{87}$$

Moreover, since $\gamma \in (0, 1]$ and $C_t(\cdot) \geq 0$, we have

$$\sum_{i=0}^{\infty} \gamma^i C_t(S_{t+1+i}) = \frac{1}{\gamma} \sum_{i=1}^{\infty} \gamma^i C_t(S_{t+i}) \leq \frac{1}{\gamma} \sum_{i=0}^{\infty} \gamma^i C_t(S_{t+i}). \tag{88}$$

Taking conditional expectation given $\mathcal{H}_t$ and recalling the definition of the SR (72) and overlap (73), we obtain

$$\mathbb{E}\left[\sum_{i=0}^{\infty} \gamma^i C_t(S_{t+1+i}) \,\Big|\, \mathcal{H}_t\right] \leq \frac{1}{\gamma} u_t(s_t, a_t). \tag{89}$$

Combining (86), (87), and (89) gives

$$\mathbb{P}(\Delta(t) > m \mid \mathcal{H}_t) \leq \frac{1}{m\,\eta\,\gamma^m} u_t(s_t, a_t). \tag{90}$$

By the uniform overlap condition (78),

$$\mathbb{P}(\Delta(t) > m \mid \mathcal{H}_t) \leq \frac{\kappa}{m\,\eta\,\gamma^m} = 1 - p. \tag{91}$$

Equivalently, at *every* time $t$,

$$\mathbb{P}(\Delta(t) \leq m \mid \mathcal{H}_t) \geq p. \tag{92}$$

**Step 2: A geometric tail bound for $\Delta_k$ in blocks of length $m$.** Fix $k \in \{1, \ldots, n-1\}$ and consider the gap $\Delta_k = \tau_{k+1} - \tau_k$. Define the events $E_j := \{\Delta_k > jm\}$ for $j \geq 0$. We claim

$$\mathbb{P}(\Delta_k > jm) \leq (1 - p)^j, \qquad j = 0, 1, 2, \ldots \tag{93}$$

We prove (93) by induction. For $j = 0$, the statement holds since the probability is 1. Assume it holds for $j$.

On the event $E_j = \{\Delta_k > jm\}$, no new state has been discovered by time $\tau_k + jm$, so the next $m$-step discovery probability from time $\tau_k + jm$ is still bounded below by (92), applied at the (random) time $t = \tau_k + jm$. Conditioning on the history at that time and then integrating over the randomness of the history, we get

$$
\begin{aligned}
\mathbb{P}(\Delta_k > (j+1)m) &= \mathbb{E}[\mathbb{I}\{E_j\}\,\mathbb{P}(\Delta_k > (j+1)m \,|\, \mathcal{H}_{\tau_k+jm})] \\
&\leq \mathbb{E}[\mathbb{I}\{E_j\}(1-p)] = (1-p)\,\mathbb{P}(E_j) \\
&\leq (1-p)\,(1-p)^j = (1-p)^{j+1},
\end{aligned}
\tag{94}
$$

which completes the induction and establishes (93).

**Step 3: Summing the tail yields $\mathbb{E}[\Delta_k] \leq m/p$.** Using the tail-sum formula for nonnegative integer-valued random variables,

$$
\begin{aligned}
\mathbb{E}[\Delta_k] &= \sum_{t=0}^{\infty} \mathbb{P}(\Delta_k > t) \\
&\leq \sum_{j=0}^{\infty} \sum_{t=jm}^{(j+1)m-1} \mathbb{P}(\Delta_k > t) \\
&\leq \sum_{j=0}^{\infty} \sum_{t=jm}^{(j+1)m-1} \mathbb{P}(\Delta_k > jm) \\
&\leq \sum_{j=0}^{\infty} m\,(1-p)^j = \frac{m}{p},
\end{aligned}
\tag{95}
$$

which is (82).

Finally, since $T_{\text{cov}} = \tau_n = \tau_1 + \sum_{k=1}^{n-1} \Delta_k$ and $\tau_1 = 0$, taking expectation gives

$$
\mathbb{E}[T_{\text{cov}}] = \sum_{k=1}^{n-1} \mathbb{E}[\Delta_k] \leq \sum_{k=1}^{n-1} \frac{m}{p} = (n-1)\frac{m}{p},
$$

which is (83). □

**Interpretation.** Proposition A.2 states that if FPVR maintains a uniform upper bound on the unnormalized future–past overlap $u_t(s_t, a_t)$ while the persistence representation does not forget previously visited states, then with a uniformly positive probability the trajectory discovers a new state within the next $m$ steps, yielding an $O(n)$ upper bound on the expected cover time. Although the assumptions are idealized, the result provides an analytical explanation for why suppressing short-horizon revisitation (low overlap) can improve coverage efficiency.

## B. DQN with FPVR Exploration

**Notation.** Let $\mathcal{A}$ denote the (fixed) action set and $\{\epsilon_t\}$ the exploration schedule. We use a replay buffer $\mathcal{D}$ storing transitions $(s_t, a_t, r_t, s_{t+1}, d_t)$, where $d_t \in \{0, 1\}$ indicates episode termination. Let $\phi$ be a frozen encoder and let $\mathcal{B}_t$ denote a buffer of recent encoder features used to estimate whitening statistics. Every $K$ environment steps, we update the ZCA whitening statistics $(\mu_t, \Sigma_t)$ via (28)–(29) (EMA rate $\rho$), and compute the whitened feature $\tilde{\phi}^{(t)}$ by (30). The whitened PF accumulator $\tilde{\chi}_t^{(t)}$ is updated by (31) and is reset at episode starts. The whitened SF head $\tilde{\psi}_\theta^{(t)}(s, a)$ is trained with target (32) and loss (33). FPVR is the cosine similarity $\tilde{\mathcal{F}}^{(t)}(s_t, a)$ defined in (26); its action-wise z-score (16) yields $\bar{\tilde{\mathcal{F}}}^{(t)}(s_t, a)$, which is injected into the biased $Q$ function via (17). Algorithm 1 provides a detailed implementation for DQN+FPVR.

## C. Experimental Setup and Implementation Details

In this paper, experiments on the tabular *MiniGrid-FourRooms* environment were conducted on an Intel i7-13700HX CPU, while experiments on the visual *MiniGrid-Maze* and *Atari 2600* games were performed on 8×Nvidia RTX 4090 GPU. The

---

**Algorithm 1** DQN with FPVR exploration (implementation-level pseudocode)

---

**Require:** fixed action set $\mathcal{A}$; replay buffer $\mathcal{D}$; feature buffer $\mathcal{B}_t$; discounts $\gamma, \lambda$; Q-bias weight $\alpha$; exploration schedule $\{\epsilon_t\}$; whitening update period $K$; EMA rate $\rho$; update period $U$.

1: Initialize a frozen encoder $\phi$ (randomly) and freeze its parameters.
2: Initialize DQN parameters for $Q$ and target network $Q^- \leftarrow Q$; initialize SF-head parameters $\theta$; initialize whitening stats $(\mu_0, \Sigma_0)$; set PF $\tilde{\chi}_0^{(0)} \leftarrow \mathbf{0}$.
3: **for** $t = 0, 1, 2, \ldots$ **do**
4:     **if** episode starts **then**
5:         $\tilde{\chi}_t^{(t)} \leftarrow \mathbf{0}$
6:     **end if**
7:     Observe $s_t$; compute encoder feature $\phi(s_t)$; push $\phi(s_t)$ into $\mathcal{B}_t$.
8:     **if** $t$ is a whitening-update step (every $K$ steps) **then**
9:         Sample a batch of features from $\mathcal{B}_t$ and compute $(\hat{\mu}_t, \hat{\Sigma}_t)$ by (28); update $(\mu_t, \Sigma_t)$ by (29).
10:     **end if**
11:     Compute $\tilde{\phi}^{(t)}(s_t)$ by (30) using the current $(\mu_t, \Sigma_t)$.
12:     Update PF online by (31) using $\tilde{\phi}^{(t)}(s_t)$.
13:     For all $a \in \mathcal{A}$, compute FPVR $\tilde{\mathcal{F}}^{(t)}(s_t, a)$ by (26) and its action-wise z-score $\bar{\tilde{\mathcal{F}}}^{(t)}(s_t, a)$ by (16).
14:     Form biased values $Q_b^{(t)}(s_t, a)$ as in (17) (with $\bar{\tilde{\mathcal{F}}}^{(t)}$ in place of $\bar{\mathcal{F}}^{(t)}$), and select $a_t$ by $\epsilon_t$-greedy on $Q_b^{(t)}(s_t, \cdot)$.
15:     Step the environment with $a_t$; observe $(r_t, s_{t+1}, d_t)$; store $(s_t, a_t, r_t, s_{t+1}, d_t)$ in $\mathcal{D}$.
16:     **if** $t$ is an update step (every $U$ steps) **then**
17:         Sample a minibatch from $\mathcal{D}$ and perform the standard DQN (MMC) update on $Q$ as in Machado et al. (2020).
18:         Using the same minibatch, update the SF head: whiten all sampled states with the current $(\mu_t, \Sigma_t)$ and then apply the SF target (32) and loss (33).
19:         Periodically update target network $Q^- \leftarrow Q$.
20:     **end if**
21: **end for**

---

detailed experimental settings for each environment and task are shown as below.

## C.1. Tabular MiniGrid-FourRooms

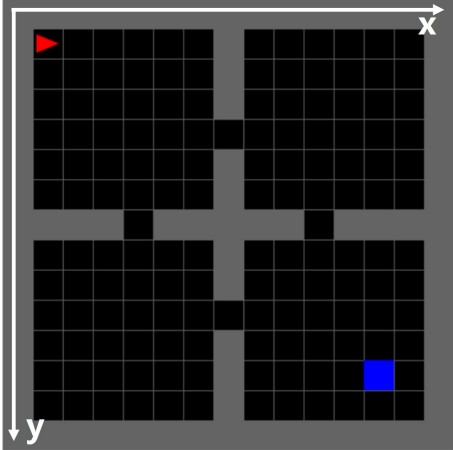

*Figure 8.* Tabular *MiniGrid-FourRooms* environment.

**Environment overview**

Tabular *MiniGrid-FourRooms* is a customized and wrapped environment in the MiniGrid framework (Chevalier-Boisvert et al., 2023). As shown in Figure 8, in a $W \times H = 15 \times 15$ grid environment (where black grids represent reachable areas, and gray grids represent walls), an agent (red triangle) performs an exploration-only/goal-reaching task.

- **Observation space:** Tabular state ID $s = yW + x$ with $W = H = 15$, so $|\mathcal{S}| = WH = 225$ (walls are included in the index but treated as blocked/self-loop in the transition table; coverage is counted over visited states during interaction).

- **Action space:** $|\mathcal{A}| = 4$ with actions $\{0 :\uparrow,\ 1 :\downarrow,\ 2 :\leftarrow,\ 3 :\rightarrow\}$.

- **Task description:** We consider two tasks in the same *MiniGrid-FourRooms* environment.

  In the **exploration-only** task, the agent is initialized at $(x, y) = (1, 1)$ each episode which includes 3,000 steps. Each exploration method is evaluated purely by exploration efficiency, i.e., how quickly it expands state coverage (number of unique visited states as a function of steps, optionally also reported with a periodically reset window).

  In the **goal-reaching** task, a goal cell (blue square in Figure 8) provides reward $+1$ upon entry and terminates the episode early, while all other steps receive no reward; at the beginning of each episode, the agent start position is sampled uniformly at random from free (non-wall) cells excluding the current goal cell. Crucially, the goal location is refreshed every 40,000 training steps (the refresh can occur mid-episode), yielding a periodically changing goal schedule that is kept identical across compared methods for the same random seed.

- **Hyperparameter settings:** Table 2 and Table 3 demonstrate the key hyperparameters of the exploration-only task and the goal-reaching task, respectively.

*Table 2.* Key hyperparameters for exploration-only *MiniGrid-FourRooms*.

| Method | Behavior policy | SR $(\gamma_{\text{sr}}, \eta_{\text{sr}})$ | $C_t$-discount $\lambda$ | SARSA $(\epsilon, \alpha, \gamma)$ |
|---|---|---|---|---|
| FPVR | Boltzmann $(\beta = 10)$ | $(0.9,\ 0.1)$ | 0.999 | – |
| $r^{FP}$+SARSA | $\epsilon$-greedy $(\epsilon = 0.1)$ | – | 0.999 | $(0.1,\ 0.1,\ 0.99)$ |
| SP+SARSA | $\epsilon$-greedy $(\epsilon = 0.1)$ | $(0.9,\ 0.1)$ | – | $(0.1,\ 0.1,\ 0.99)$ |
| SR+SARSA | $\epsilon$-greedy $(\epsilon = 0.1)$ | $(0.9,\ 0.1)$ | – | $(0.1,\ 0.1,\ 0.99)$ |
| Random Walk | uniform random | – | – | – |

*Table 3.* Key hyperparameters for goal-reaching *MiniGrid-FourRooms*. The intrinsic-reward coefficient and the FPVR value-function bias coefficient were selected via a grid search over $\{0.1, 0.01, 0.001, 0.0001\}$, choosing the value that yielded the best performance.

| Method | Behavior policy | SR $(\gamma_{\text{sr}}, \eta_{\text{sr}})$ | $C_t$-discount $\lambda$ | SARSA $(\epsilon, \alpha, \gamma)$ | Intrinsic reward coeff. |
|---|---|---|---|---|---|
| FPVR+SARSA | $\epsilon$-greedy $(\epsilon = 0.1)$ on $Q_b$ $(\alpha_{\text{FPVR}} = 0.01)$ | $(0.9,\ 0.1)$ | 0.999 | $(0.1,\ 0.1,\ 0.9)$ | – |
| $r^{FP}$+SARSA | $\epsilon$-greedy $(\epsilon = 0.1)$ | – | 0.999 | $(0.1,\ 0.1,\ 0.9)$ | $\beta_{\text{rFP}} = 0.001$ |
| SP+SARSA | $\epsilon$-greedy $(\epsilon = 0.1)$ | $(0.9,\ 0.1)$ | – | $(0.1,\ 0.1,\ 0.9)$ | $\beta_{\text{SP}} = 10^{-4}$ |
| SR+SARSA | $\epsilon$-greedy $(\epsilon = 0.1)$ | $(0.9,\ 0.1)$ | – | $(0.1,\ 0.1,\ 0.9)$ | $\beta_{\text{SR}} = 10^{-4}$ |
| SARSA | $\epsilon$-greedy $(\epsilon = 0.1)$ | – | – | $(0.1,\ 0.1,\ 0.9)$ | – |

## C.2. Visual MiniGrid-Maze

### Environment overview

Visual *MiniGrid-Maze* is a MiniGrid-based maze environment that provides a rendered RGB image observation. As shown in Figure 9, the agent navigates in a $W \times H = 20 \times 20$ grid with fixed walls (black reachable cells versus gray walls), under an exploration-only setting with no extrinsic reward.

- **Observation space:** At each step, the environment returns an RGB image of the current maze configuration (rendered frame). In our implementations, the image is converted to grayscale and resized to $128 \times 128$ before being fed into the learning algorithm, resulting in a single-channel visual observation.

- **Action space:** $|\mathcal{A}| = 4$ with actions $\{0 :\uparrow,\ 1 :\downarrow,\ 2 :\leftarrow,\ 3 :\rightarrow\}$, i.e., global moves on the grid (independent of the agent's orientation); moves into walls result in no position change.

- **Task description:** We consider an **exploration-only** task with no extrinsic reward. Each method interacts with the maze for 9,000 steps and is evaluated by (i) cumulative coverage (the number of unique visited positions versus steps), (ii) windowed coverage with periodic reset every 3,000 steps for visualization, and (iii) visitation-count heatmaps over $(x, y)$ (ignoring orientation).

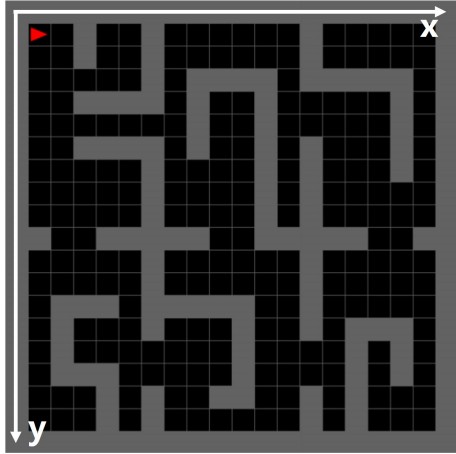

Figure 9. Visual *MiniGrid-Maze* environment.

- **Hyperparameter settings and network-architecture:** Table 4 summarizes the key hyperparameters used in this visual exploration-only task. **Behavior policies:** In the two FPVR baselines, actions are sampled from a Boltzmann distribution over $\bar{\mathcal{F}}^{\pi}(s_t, a)$ with temperature parameter $\beta = 1.0$. For the two DQN-based baselines (SR+DQN and SP+DQN), actions are selected using an $\epsilon$-greedy policy where $\epsilon$ is annealed linearly from 1.0 to 0.1 over the first 1,000 interaction steps. The random-walk baseline chooses actions uniformly at random. Figure 10 provides the network-architecture adopted by the FPVR exploration.

Table 4. Key hyperparameters for exploration-only tasks in visual *MiniGrid-Maze*.

| Method | (SR,PR) Discount $(\gamma_{\text{sr}}, \lambda)$ | DQN discount $\gamma_q$ | Buffer capacity, batch size | SR learning rate | DQN learning rate |
|---|---|---|---|---|---|
| FPVR (deep) | (0.9, 0.95) | – | 3,000, 64 | $10^{-3}$ | – |
| FPVR (tabular) | (0.9, 0.999) | – | – | 0.1 | – |
| SR + DQN | $\gamma_{\text{sf}} = 0.9$ | 0.99 | 3,000, 64 | $2.5 \times 10^{-4}$ | $2.5 \times 10^{-4}$ |
| SP + DQN | $\gamma_{\text{sf}} = 0.9$ | 0.99 | 3,000, 64 | $2.5 \times 10^{-4}$ | $2.5 \times 10^{-4}$ |

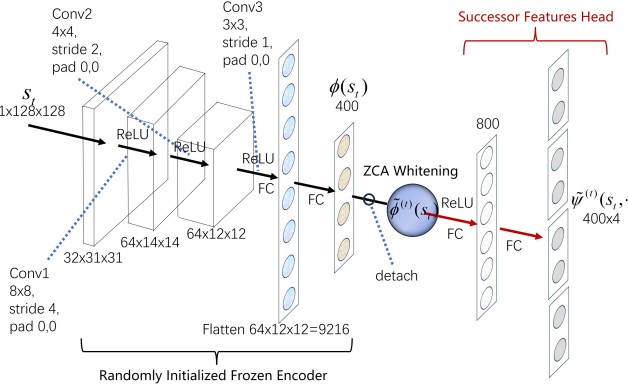

Figure 10. Network structure for whitened successor features in visual *MiniGrid-Maze*.

## C.3. Atari 2600 games

### Environment overview

To examine the ability of FPVR to discover novel states in high-dimensional, continuous state spaces, we consider six hard-to-explore *Atari 2600* games from the Arcade Learning Environment (ALE) (Bellemare et al., 2013), as shown in Figure 11. The results reported in Table 1 are the average score of 10 policies obtained after training for 100M frames with

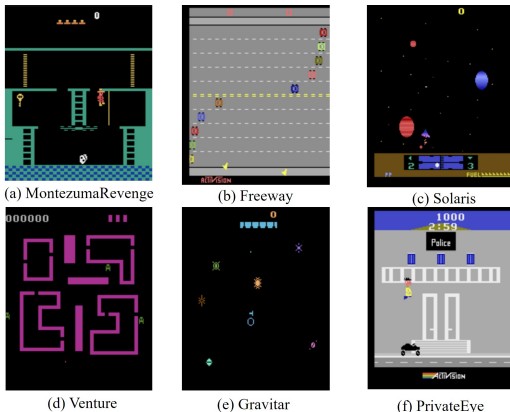

(a) MontezumaRevenge  (b) Freeway  (c) Solaris

(d) Venture  (e) Gravitar  (f) PrivateEye

*Figure 11.* Hard-to-explore *Atari 2600* games environment.

*Table 5.* Key hyperparameter settings for DQN in *Atari 2600* games.

| Hyperparameter | Value |
|---|---|
| Minibatch size | 32 |
| Replay buffer capacity | 1,000,000 |
| Frame stack | 4 |
| Target network update frequency | 40,000 |
| $Q$ Discount factor | 0.99 |
| Frame skip | 5 |
| Update frequency | 4 |
| Learning rate (Q network) | $2.5 \times 10^{-4}$ |
| RMSprop momentum | 0.0 |
| RMSprop squared-gradient momentum | 0.95 |
| RMSprop $\epsilon$ | 0.01 |
| Initial exploration probability | 1.0 |
| Final exploration probability | 0.1 |
| Final exploration frame | 1,000,000 |
| No-op max | 30 |
| Sticky action probability | 0.25 |
| Reward clipping | $\mathrm{clip}([-1, 1])$ |
| Full action space | False |
| Mixed Monte Carlo return weight | 0.5 |

10 different random seeds. Each trained policy is evaluated for 30 trials using an $\epsilon$-greedy policy with $\epsilon = 0.05$, and its score is defined as the mean return over these runs.

- **Observation space:** At each step, the environment returns an RGB image of the current frame. In our implementation, the image is converted to grayscale and resized to $84 \times 84$ as in Mnih et al. (2015).

- **Action space:** The full discrete *Atari 2600* action set contains 18 actions. The effective action set of each game is a subset of these 18 actions. To match the open-source implementation of Machado et al. (2020), we use the game-dependent minimal action set required by each game.

- **Hyperparameter settings and network-architecture:** To align with the settings of Machado et al. (2020), the hyperparameters in Table 5 are adopted. Main hyperparameters about the training of the whitened SF network is illustrated in Table 6. The structure of the SF network adopted by the FPVR method is shown in Figure 12. Note that, for simplicity, our implementation does not use a target network to update the whitened SF network. To ensure a fair comparison, the $Q-$network used in FPVR+DQN$^{\mathrm{MMC}}$ is obtained by removing the reconstruction head and the SF head from the network architecture adopted by Machado et al. (2020), keeping only the remaining components.

*Table 6.* Key hyperparameter settings for FPVR exploration in *Atari 2600* games.

| Hyperparameter | Value |
| --- | --- |
| FPVR network learning rate | $5 \times 10^{-4}$ |
| Successor features discount $\gamma_{\mathrm{sr}}$ | 0.5 |
| Persistence features discount $\lambda$ | 0.9 |
| Whitening update period | 1000 |
| Whitening EMA learning rate | 0.001 |
| Covariance buffer size | 10,000 |
| Reset $\tilde{\chi}_t$ each episode | True |
| FPVR weight for biased $Q$ $\alpha$ | 0.001 |

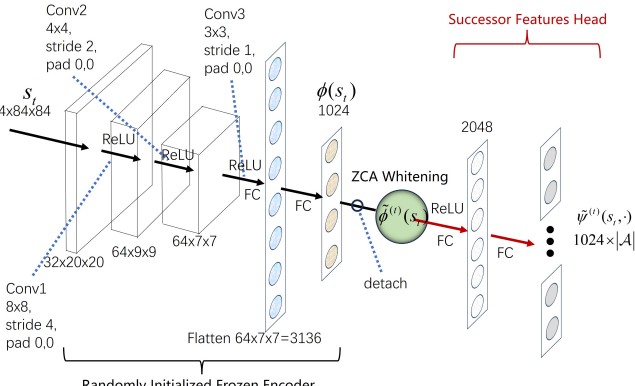

*Figure 12.* Network structure for whitened successor features in *Atari 2600* games.

## D. Supplementary Experimental Results

FPVR exploration is built upon the dual-timescale state representation (DTSR). While the form of the persistence representation (PR) is relatively fixed, the successor representation (SR) admits multiple variants, depending on whether it continues to be updated and how the TD target is selected. In this section, we present additional experiments to provide a primary understanding of how the SR learning influences the effectiveness of FPVR exploration.

### D.1. FPVR with a well-learned successor representation

Novelty-based intrinsic rewards may vanish as exploration and learning proceed (Jarrett et al., 2023; Castanyer et al., 2024), and thus fail to provide persistent guidance for exploration. We consider the exploration-only task shown in Figure 8. To investigate whether FPVR's strong exploration performance relies on the dynamic update of the SR, or whether a fixed SR combined with a dynamically updated PR is sufficient to achieve efficient exploration in environments with stationary transition dynamics, we first let the agent perform a random walk for 30,000 steps and train the SR on-policy. We then freeze the SR and update only the PR, executing FPVR based on a fixed SR and a dynamic PR (all other settings follow the FPVR configuration used in the tabular *MiniGrid-Maze* exploration-only experiment).

As shown in Figure 13, FPVR with a well-learned SR and an online-updated PR maintains high coverage efficiency from the very beginning of the exploration (even higher than the standard FPVR baseline). This suggests that FPVR does not rely on the SR updating process itself; rather, an accurate SR prior can further improve FPVR's early-stage exploration efficiency.

To further demonstrate the dynamics of state coverage under FPVR with a fixed SR, Figures 14, 15, and 16 show the average state visitation maps (over 50 seeds) at 100, 500, and 3,000 time steps in the exploration-only task for different exploration methods. The results indicate that although FPVR with a fixed SR achieves high coverage efficiency from the start, its visit-count distribution becomes less uniform as exploration proceeds compared to standard FPVR, $r^{FP}$+SARSA or SP+SARSA, exhibiting relatively higher visitation near walls that impede transitions.

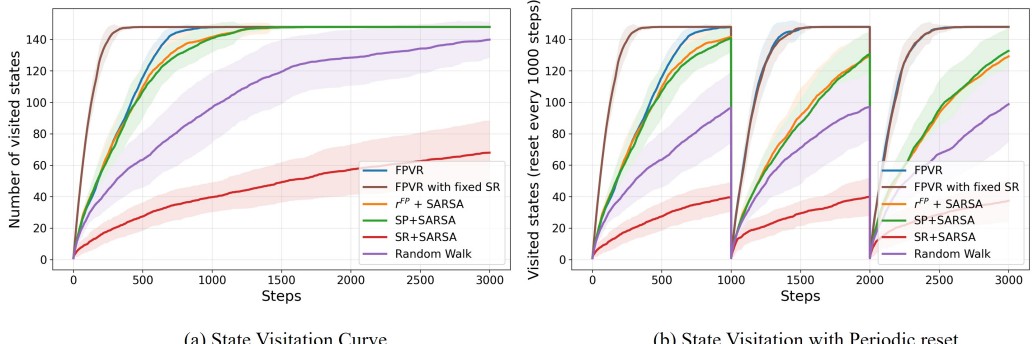

(a) State Visitation Curve  (b) State Visitation with Periodic reset

*Figure 13.* Coverage efficiency validation in tabular *MiniGrid-FourRooms* considering FPVR with fixed SR.

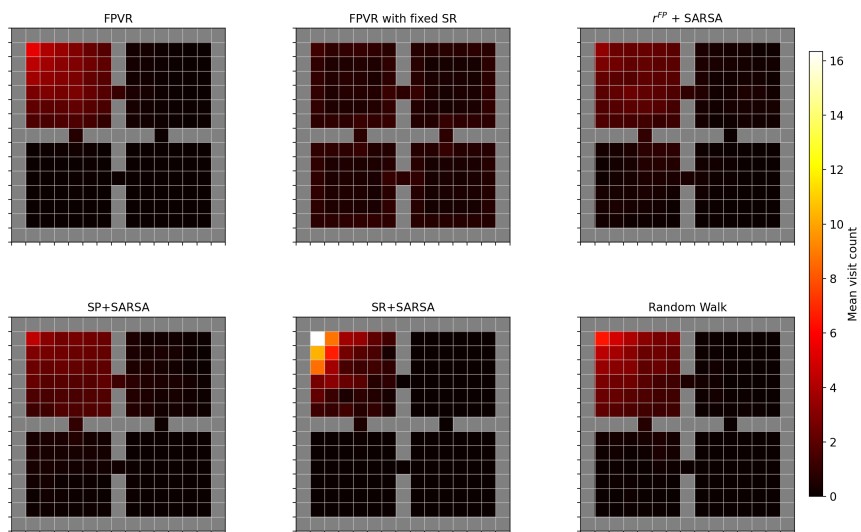

*Figure 14.* Average visit-count distribution in *MiniGrid-FourRooms* exploration-only task over 100 steps.

## D.2. Different update targets of the successor representation

As introduced in Section 2, the successor representation (SR) admits multiple variants depending on how the bootstrap action is selected when forming the TD target. Here we compare FPVR exploration under three SR-target choices: (i) **current/on-policy** (SR target=current): the bootstrap action is sampled from the current behavior policy; (ii) **min-FPVR** (SR target=min): the bootstrap action is chosen as the action minimizing the FPVR score, i.e., $\arg\min_a \tilde{\mathcal{F}}^{(t)}(s_t, a)$; (iii) **uniform-random/mean** (SR target=mean): the bootstrap action is sampled uniformly at random over the action set.

In both tabular *MiniGrid-FourRooms* and visual *MiniGrid-Maze*, we keep the experimental setup and FPVR configuration identical to the main experiments, and only change the SR-target choice. The tabular results indicate that the three SR-target variants achieve similar coverage efficiency, while in the visual setting the min and mean targets yield slightly higher coverage efficiency than current.

We hypothesize two possible explanations. First, the min target resembles a policy-evaluation / policy-improvement alternation (analogous to Q-learning-style greedy bootstrapping): although the SR coefficients within the unnormalized FPVR overlap are non-stationary, the min target can still bias SR learning toward visitation predictions that reduce future–past overlap. Second, the mean target may improve coverage by decoupling SR learning from the evolving behavior policy, allowing SR to capture a more stable, averaged characterization of environment dynamics; FPVR computed from such a policy-agnostic SR can then more reliably identify actions that reduce overlap on average.

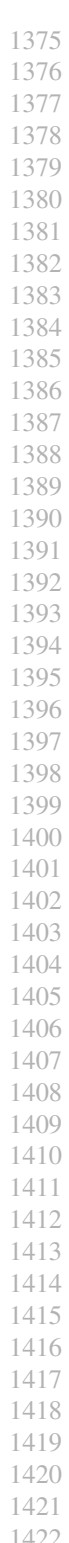

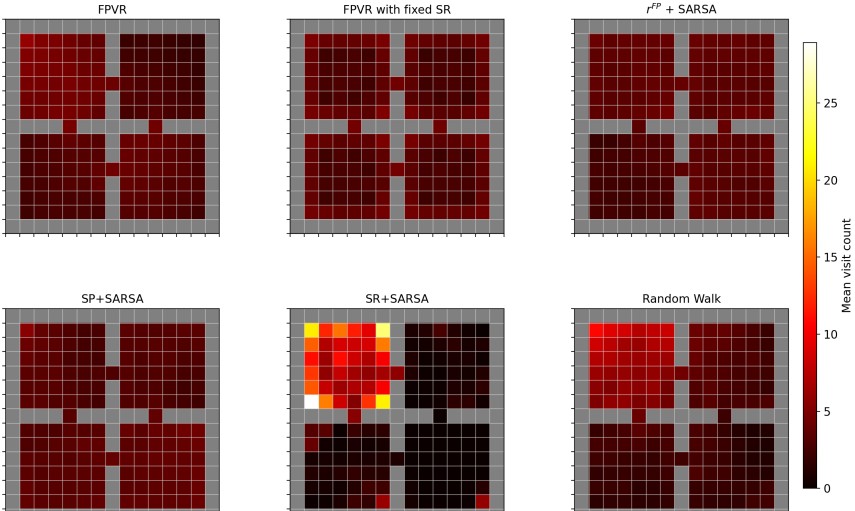

*Figure 15.* Average visit-count distribution in *MiniGrid-FourRooms* exploration-only task over 500 steps.

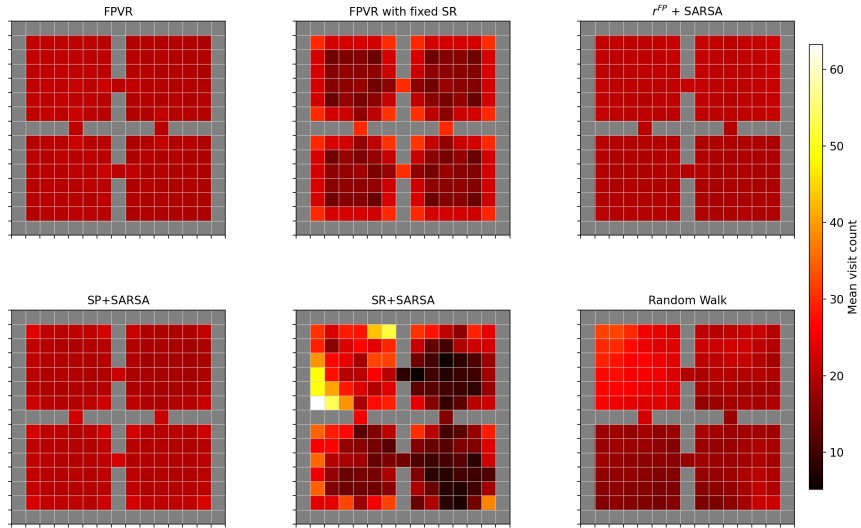

*Figure 16.* Average visit-count distribution in *MiniGrid-FourRooms* exploration-only task over 3000 steps.

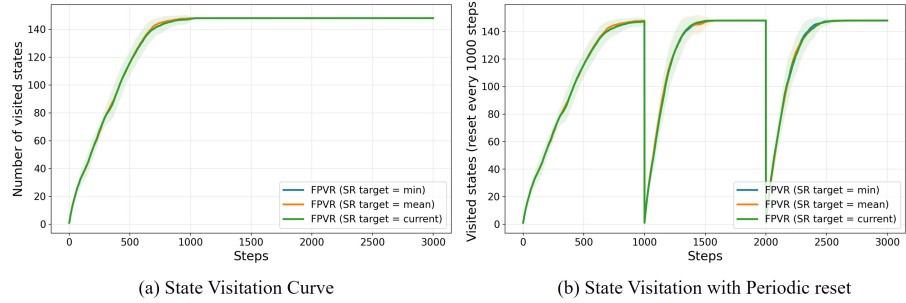

(a) State Visitation Curve       (b) State Visitation with Periodic reset

*Figure 17.* Average visit-count distribution in *MiniGrid-FourRooms* exploration-only task over 3000 steps.

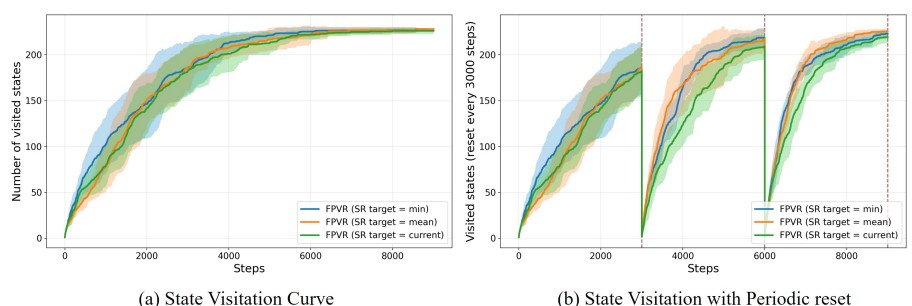

(a) State Visitation Curve     (b) State Visitation with Periodic reset

*Figure 18.* Comparison between FPVR exploration with different SR targets in visual *MiniGrid-Maze*.

