# OpenReview forum: "Future–Past Visitation Redundancy for Rapid Coverage Exploration in Reinforcement Learning"
_ICML.cc/2026/Conference — Submitted to ICML 2026_

### Official Review · Reviewer_J6Qy · 2026-02-18

**Soundness:** 3
**Presentation:** 3
**Significance:** 2
**Originality:** 3
**Overall Recommendation:** 4
**Confidence:** 4

**Summary:**

To consider new state exploration and coverage speed simultaneously, this paper proposes Future–Past Visitation Redundancy (FPVR) for
exploration in Reinforcement Learning. The key to FPVR is Dual-Time scale State Representation (DTSR), which not only records an online discounted accumulation of past state features but also encodes the expected discounted accumulation of future state features. The exploration in FPVR is to suppress short-term revisitation to improve coverage speed while biasing exploration toward new states. The experimental study demonstrates improved performance.

**Compliance With Llm Reviewing Policy:**

Affirmed.

**Final Justification:**

Thanks for your response. I will keep my score.

**Key Questions For Authors:**

1) The definition of successor features seems somewhat redundant. Can it be directly extended by the Q function？
2) Can authors visually show the difference between $a_s$ and $b_s$ in an environment？

**Limitations:**

The authors should discuss the limitations in the Conclusion.

**Strengths And Weaknesses:**

Strengths :
1) The paper is generally well-organized, making it easy for readers to follow.
2) The overall idea is novel but sound. The proposed method is reasonable from an intuitive perspective.
3) The proposed method shows decent performance in the experiments, and the effectiveness of FPVR is numerically illustrated.

Weakness:
1) Compared to existing methods, the proposed method seems to perform decently in the numerical experiments, but the improvement is not that significant. Without additional analysis on the quality of the learned representation, it is not clear if the performance benefits indeed come from FPVR.
2) Entropy-based exploration, e.g., SAC, can be compared with the FPVR.
3) For some real-world reinforcement learning tasks, the assumption of $r_t=r (s_t)$ is too strict.

---

> ### Author Rebuttal · Authors · 2026-03-31
>
> Dear Reviewer J6Qy,
>
> Thank you for the thoughtful review and encouraging feedback. We appreciate your recognition of the clarity, intuition, and empirical effectiveness of FPVR. Below we respond to your key questions and noted weaknesses.
>
> ## **1. Key Questions**
>
> ### **Q1. The definition of successor features seems somewhat redundant. Can it be directly extended by the Q function?**
>
> Thank you for this insightful question. We agree that this is a natural observation. Indeed, the final FPVR quantity can be written in a normalized Q-function-like form, and the paper already shows that the unnormalized FPVR is equivalent to the Q-function of a **history-dependent intrinsic reward** (Proposition 3.3 and the appendix discussion).
>
> However, FPVR is still different from directly learning such a Q-function. First, **SR/SF captures environment dynamics rather than a specific reward**, so when rewards change in the same environment, a learned SR can still support efficient early exploration. Second, in **continuous or high-dimensional state spaces**, SF/PF avoids relying on explicit state-visit counting. Third, SR/SF lets the history-dependent exploration signal be **propagated much more immediately** than repeatedly relearning a non-stationary intrinsic-reward value function. Thus, while FPVR admits an equivalent Q-function interpretation, its mechanism and practical advantages are not redundant with directly learning a standard Q-function.
>
> ### **Q2. Can authors visually show the difference between $a_s$ and $b_s$ in an environment?**
>
> Thank you for this helpful suggestion. In Eq. 14, $a_s$ and $b_s$ denote the state-index coefficients of SR and PR in the tabular case, i.e., the occupancy weights assigned to each state by future and past visitation. Visualizing them indeed helps explain the design intuition of FPVR.
>
> The original Figure 1 provides a conceptual illustration in MontezumaRevenge, but it is not a precise quantitative visualization. To make this more explicit, we additionally plotted the SR and PR distributions in **MiniGrid-FourRooms** at a particular time step during exploration, where SR is taken after sufficient iterations and is approximately converged. The results are provided here: [SR/PR distribution visualization](https://anonymous.4open.science/r/distribution_sr_pr-184D). We will incorporate this analysis in the revision.
>
> ## **2. Weaknesses**
>
> ### **W1. Limited analysis on the learned representation quality**
>
> Thank you for highlighting this important issue. We agree that further analysis of the learned representation would strengthen the paper. To address this, we added the tabular SR/PR distribution visualization above: [SR/PR distribution visualization](https://anonymous.4open.science/r/distribution_sr_pr-184D). For higher-dimensional and continuous-state settings, more detailed visualization of learned SR/PR representations is also an important direction, and we will emphasize this more clearly in the revision and future work.
>
> ### **W2. Entropy-based exploration (e.g., SAC) should be compared**
>
> Thank you for this important suggestion. The main reason we did not originally compare with SAC-style entropy exploration is that the FPVR method in the paper had not yet been extended to **continuous action spaces**. To address this, we have now developed an **actor-critic form of FPVR** that can be applied to continuous-action tasks; the detailed design is given in our response to **Reviewer dsZX, Q3**. Using this extension, we compared FPVR against SAC-style entropy-driven exploration on the **PointMazeLarge-v3** pure exploration task, focusing only on state-space coverage efficiency. The results, available here: [PointMaze FPVR results](https://anonymous.4open.science/r/point_maze_fpvr-95EC), show that FPVR achieves better coverage efficiency than entropy-driven exploration in this setting. We will include these results in the revision.
>
> ### **W3. The assumption on reward representation is too strict for some real-world tasks**
>
> Thank you for pointing out this limitation. We agree that the exact assumption used for strict value-function decomposition can be restrictive in some real-world tasks. However, this does not mean that SR/SF becomes unusable once the assumption is violated; rather, the value representation becomes approximate instead of exact. This assumption is standard in the SR/SF literature and has been widely used [1-3]. We will further clarify this point in the revision.
>
> ## **References**
>
> [1] P. Dayan, “Improving Generalization for Temporal Difference Learning: The Successor Representation,” *Neural Computation*, 1993.
> [2] A. Barreto *et al*., “Successor Features for Transfer in Reinforcement Learning,” *NeurIPS*, 2017.
> [3] M. C. Machado, M. G. Bellemare, and M. Bowling, “Count-Based Exploration with the Successor Representation,” *AAAI*, 2020.

---

> > ### Author Rebuttal · Reviewer_J6Qy · 2026-04-03
> >
> > Thank you for your reply. Most of my concerns have been addressed.

---

> > > ### Author Response · Authors · 2026-04-04
> > >
> > > Dear Reviewer J6Qy,
> > >
> > > Thank you very much for your follow-up and for taking the time to review our rebuttal. We are pleased that our additional clarifications and results helped address your remaining concerns.
> > >
> > > We also appreciate your constructive suggestions, which have been valuable in improving the paper's presentation and strengthening its experimental discussion.
> > >
> > > Thank you again for your time and consideration. Wishing you all the best.

---

### Official Review · Reviewer_dsZX · 2026-03-06

**Soundness:** 2
**Presentation:** 3
**Significance:** 3
**Originality:** 3
**Overall Recommendation:** 4
**Confidence:** 3

**Summary:**

This work introduces an exploration mechanism for reinforcement learning that aims to improve coverage efficiency in sparse-reward environments. The main idea is to discourage actions where the predicted future visitation patterns overlap strongly with states that have been recently visited. To do this, they use successor representations to capture expected discounted future state occupancy and a persistence representation to summarise recently visited states along the trajectories. The overlap of these representations is normalized and subtracted from the Q function to penalize such actions.They provide a theoretical analysis in the tabular case that connects this penalty to an intrinsic reward interpretation and offer empirical results on MiniGrid and Atari tasks that indicate improved coverage efficiency.

**Compliance With Llm Reviewing Policy:**

Affirmed.

**Final Justification:**

The authors' responses have addressed my main concerns and so I have increased my score to a weak accept. The paper is well written, appears to be sound although I did not check all the details in the appendix, and results make this a significant contribution.

**Key Questions For Authors:**

1. How robust is the approach when the successor representation is poorly learned early in training?
2. What is the contribution of each component (SR, persistence representation, whitening, normalization)? For example, how does performance change if whitening is removed or if only the persistence signal is used?
3. Can additional experiments on continuous state-action spaces be added? Also perhaps more modern exploration methods designed for long horizon sparse exploration

**Limitations:**

Yes

**Strengths And Weaknesses:**

Strengths:
- The main idea is that exploration is inefficient when the agent's predicted future states overlap with recently visited states. This idea is reasonable and the technical implementation is also nice.
- The paper is generally well structured and the high level approach is very clear.
- Efficient exploration remains a central challenge in RL, particularly in sparse reward environments and such a lightweight exploration mechanism that improves coverage without introducing complex auxiliary objectives is practically useful. This approach is specifically targeted towards value-based agents because it integrates directly into action selection.
- Both successor representations and visitation-based exploration signals have been studied previously, but the idea to combine a long timescale predictive visitation model with a short timescale persistence summary seems original. This is different from the common curiosity based approaches that optimize for novelty maximization explicitly while this work frames exploration as suppression of redundant visitations.

Weaknesses:
- Successor representation learning is unreliable early on which is often when exploration matters most. An experiment showing the learning curves vs successor representation accuracy would be valuable to see.
- An ablation on the representations such as raw features vs whitened features vs normalized features would also be a useful addition.
- The cover-time bound in Proposition A.2 is appears circular to me. The key assumption requires that FPVR maintains a uniform upper bound on the future-past overlap u_t(s_t, a_t) ≤ κ along the entire trajectory. But this is precisely the property that FPVR is designed to induce. The theorem assumes this condition and derives consequences. This is not a strong theoretical contribution and moreover seems disconnected from the empirical setting with function approximation.
- It is mentioned that this approach scales to continuous state-action spaces however the experiments are limited to tabular and discrete settings.

---

> ### Author Rebuttal · Authors · 2026-03-31
>
> Dear Reviewer dsZX,
>
> Thank you for the thoughtful review and helpful suggestions. Below we respond to your questions and noted weaknesses.
>
> ## **1. Key Questions**
>
> ### **Q1. Early SR accuracy**
>
> Thank you for this thoughtful question. FPVR does not require SR/SF to be highly accurate early on; it mainly relies on relative action differences, and even a crude SR can already bias the agent away from recently visited states. Empirically, a well-learned SR further strengthens FPVR, while the main results already show that FPVR trained from scratch achieves higher early exploration efficiency than the baselines. For details, please see our response to **Reviewer 9kc2, Q2** above.
>
> ### **Q2. Component contributions**
>
> Thank you for raising this important question. The roles are distinct: **SR/SF** predicts future visitation and propagates the exploration signal; **PR/PF** summarizes recent visitation; **whitening** suppresses feature co-activation / scale / correlation effects; and **action-wise z-score normalization** amplifies small FPVR differences across actions.
>
> Regarding “only persistence signal”: PR/PF alone cannot define an action-conditioned redundancy score because it has no future predictor. Under the value-function interpretation, SR/SF must be paired with reward weights to define an action value, whereas the persistence signal is only a history-dependent reward weighting, so it cannot guide exploration by itself. We also added an Atari ablation comparing FPVR with and without whitening; the new row is shown below and will be filled in the revision.
>
> | Method | Freeway | Gravitar | MontezumaRevenge | PrivateEye | Solaris | Venture |
> |---|---:|---:|---:|---:|---:|---:|
> | DQN | **32.4(0.3)**| 118.5(22.0) | 0.0(0.0) | **1447.4(2567.9)** | 783.4(55.3) | 4.4(5.4) |
> | DQNMMC | 29.5(0.1) | 1078.3(254.1) | 0.0(0.0) | 113.4(42.3) | 2244.6(378.8) | 1220.1(51.0) |
> | FPVR + DQNMMC | 30.9(0.1) | **1328.8(119.6)** | **1660.0(1028.8)** | 88.7(34.0) | **2596.7(301.3)** | **1384.3(119.8)** |
> | **FPVR (no whitening) + DQNMMC** | 28.3(0.3) | 1202.0(416.0) | 0.0(0.0) | 76.9(29.0) | 1504.7(264.2) | 344.0(90.7) |
>
> ### **Q3. Continuous-action experiments**
>
> Thank you for raising this important point. The original paper mainly introduces the **FPVR concept** and validates it in tabular and continuous-state settings. To test whether FPVR extends to continuous **action** spaces, we designed a simple **actor-critic FPVR** for **pure exploration**, i.e., without extrinsic reward and focusing only on coverage efficiency.
>
> For simplicity, below $c_t$, $\phi(s_t)$, and $\psi_\theta(s,a)$ are understood to be defined in the current whitening-conditioned feature space. We maintain
> $c_t=\lambda c_{t-1}+\phi(s_t)$,
> learn a successor feature with target
> $y_\psi=\phi(s_t)+\gamma(1-d_t)\psi_{\bar\theta}(s_{t+1},a_{t+1})$,
> where $a_{t+1}\sim\pi_\eta(\cdot\mid s_{t+1},c_{t+1})$, and define
> $Q_F(s_t,a_t,c_t)=-\cos(\psi_\theta(s_t,a_t),c_t)$.
> The actor is
> $a_t\sim\pi_\eta(\cdot\mid s_t,c_t)$,
> trained with
> $L_\pi=E[\alpha\log\pi_\eta(a_t\mid s_t,c_t)-Q_F(s_t,a_t,c_t)]$.
> The principle is unchanged: **choose actions whose predicted future visitation overlaps as little as possible with the past visitation summary**.
>
> Using this extension, we conducted a pure-exploration experiment on **PointMazeLarge-v3** with **no extrinsic reward**, measuring $(x,y)$-space coverage efficiency. Besides SR and SP, we also compared against **ETD** [1]. The results are available here: [PointMaze FPVR results](https://anonymous.4open.science/r/point_maze_fpvr-95EC), and show that FPVR achieves higher coverage efficiency than the baselines. We will include these results in the revision.
>
> ## **2. Weaknesses**
>
> ### **W1, W2 & W4**
>
> Thank you for highlighting these important issues. Please see **Q1**, **Q2**, and **Q3**, respectively.
>
> ### **W3. Proposition A.2**
>
> Thank you for raising this important concern. We agree that Proposition A.2 should be viewed as an **idealized explanatory result**, not as a theorem showing that FPVR automatically guarantees a uniform bound on future–past overlap. At present, we do **not** have a strict proof of such a bound; this would likely require a more involved analysis, especially under function approximation.
>
> Its role is therefore conditional: it shows that **if** FPVR can suppress future–past overlap, **then** the cover time can be reduced. Our current reason for expecting this is algorithmic rather than theorem-based: FPVR is explicitly designed to prefer policies that minimize the overlap defined by the whitened cosine similarity. We will revise the wording to make this limited scope more explicit.
>
> ### **Reference**
>
> [1] Y. Jiang *et al*., “Episodic Novelty Through Temporal Distance,” in *ICLR*, 2025.

---

> > ### Author Rebuttal · Reviewer_dsZX · 2026-04-03
> >
> > Thank you for the response. Most of my concerns have been addressed and I am happy to increase my score.

---

> > > ### Author Response · Authors · 2026-04-04
> > >
> > > Dear Reviewer dsZX,
> > >
> > > Thank you very much for your follow-up and for your positive response to our rebuttal. We are glad that our clarifications and additional results have addressed most of your concerns.
> > >
> > > We also sincerely appreciate your helpful comments, which have helped us improve both the presentation and the empirical support of the paper.
> > >
> > > Wishing you all the best.

---

### Official Review · Reviewer_LNQk · 2026-03-11

**Soundness:** 3
**Presentation:** 3
**Significance:** 2
**Originality:** 2
**Overall Recommendation:** 3
**Confidence:** 4

**Summary:**

This paper proposes a new exploration method in RL, Future–Past Visitation Redundancy (FPVR), aiming to improve state-space coverage efficiency in sparse and non-stationary reward environments. The main idea is using the dual-timescale state representation: Successor Representation (SR) and Successor Features (SF) to capture expected discounted accumulation of future state features; using Persistence Representation (PR) and Persistence Features (PF) to capture discounted cumulative features of past states, then combining the overlap of the two into the Q-function as an action-selection bias to suppress repeated visitation and promote exploration. The paper provides mathematical derivation in the tabular setting and a high-dimensional function approximation implementation, with experiments on MiniGrid and Atari 2600 sparse-reward environments showing that FPVR improves coverage efficiency and exploration performance.

**Compliance With Llm Reviewing Policy:**

Affirmed.

**Final Justification:**

The rebuttal improves the paper’s positioning, but it does not fully resolve my main concerns, so I maintain my weak reject recommendation. In particular, the paper still does not convincingly establish that suppressing short-term future–past overlap is well aligned with useful exploration rather than simply discouraging revisitation, especially in long-horizon settings where revisiting bottleneck states can be essential. The response also continues to rely on coverage efficiency as the main justification without fully explaining when faster coverage should translate into more reward-relevant exploration. In addition, FPVR remains strongly representation-dependent in high-dimensional settings, and the rebuttal explicitly acknowledges that whitening does not resolve aliasing or representation error. Finally, key issues such as continuous-control generalization, hyperparameter sensitivity, and comparison to history-dependent intrinsic reward methods remain insufficiently supported in the submitted paper. Overall, while the clarification is appreciated, the core conceptual and empirical concerns remain.

**Key Questions For Authors:**

- In high-dimensional continuous states, does the cosine similarity of FPVR reliably reflect future–past visitation overlap? Is there theoretical support or analysis?

- How sensitive are the results to Q-bias coefficient α and PF/SF decay λ? Do these require environment-specific tuning?

- How does FPVR performance compare with history-dependent intrinsic reward methods (episodic memory, hindsight reward)?

- Does FPVR remain effective in continuous control or high-dimensional robotic tasks? Can you test the results?

**Limitations:**

Theoretical guarantees are insufficient in high-dimensional continuous spaces.

**Strengths And Weaknesses:**

## Strengths

- The motivation is clear, which identifies limitations of novelty-based intrinsic reward methods in coverage efficiency and proposes a lightweight solution.

- Provides idealized analysis of coverage time upper bound, explaining theoretically why coverage can be accelerated.

- The proposed method is easy to integrate with other algorithms. FPVR is introduced as a Q-value bias without requiring complex sub-policies or additional networks, and can be combined with value-based RL methods like DQN, SAC, and DDPG.

## Weaknesses

- In continuous/high-dimensional environments, FPVR uses cosine similarity of SF/PF to measure overlap, but there is no theoretical proof that it still guarantees coverage efficiency or convergence. Whitening can mitigate feature co-activation but cannot strictly guarantee performance.

- Key hyperparameters such as Q-bias coefficient α, PF/SF decay λ, and z-score normalization lack analysis, potentially leading to unstable performance across environments.

- Experiments are only conducted on MiniGrid and Atari games, there is no evaluation in continuous control or high-dimensional robotics tasks, limiting generalization.

- Ablation studies comparing FPVR with episodic memory or hindsight intrinsic reward methods are missing, limiting explanation of FPVR’s performance improvements.

---

> ### Author Rebuttal · Authors · 2026-03-30
>
> Dear Reviewer LNQk,
>
> Thank you for your careful reading and constructive feedback. Below we respond to your key questions and noted weaknesses.
>
> ## **1. Key Questions**
> ### **Q1. In high-dimensional continuous states, does cosine similarity reliably reflect future–past visitation overlap? Is there theoretical support?**
>
> Thank you for raising this important question. We should first clarify what “overlap” means. In RL, there is no single canonical definition of overlap between state sets or visitation patterns: for probability distributions, one may use KL divergence or cross-entropy; for vector representations, one may use cosine similarity or inner products. Therefore, in continuous/high-dimensional spaces, it is difficult to claim that one specific metric is uniquely correct.
>
> Our contribution is to propose an **effective overlap measure for improving coverage efficiency**. We do not claim that whitened cosine similarity is the only valid definition, but that it is a **practically effective overlap proxy** for exploration. In the tabular case, with one-hot basis, SR and PR can be written as $m(s_t,a)=\sum_s a_s e_s$ and $c_t=\sum_s b_s e_s$, so $\langle m(s_t,a),c_t\rangle=\sum_s a_s b_s$ , i.e., only matched state indices contribute. In continuous feature spaces, a raw SF/PF inner product contains undesired cross-terms from feature co-activation across unrelated states. Whitening suppresses these scale/correlation effects, so cosine similarity in the whitened space better approximates the tabular matched-overlap form.
>
> Thus, we do **not** claim a strict continuous-space theorem matching the tabular case. Rather, whitening plus cosine similarity provides a principled normalized surrogate, and the consistent gains in coverage efficiency serve as direct evidence of its usefulness. We will clarify this more explicitly in the revision.
>
> ### **Q2. How sensitive are the results to the Q-bias coefficient $\alpha$ and PF/SF decay $\lambda$?**
>
> Thank you for highlighting this important issue. The roles of these hyperparameters are interpretable: $\alpha$ controls exploration strength, while $\lambda$ controls the timescale of future/past accumulation.
>
> From our experiments, high coverage efficiency usually requires that the SF/PF decay should **not** be too large. In practice, values around **0.5--0.95** work better, whereas using $\lambda$ as large as a standard RL discount factor (e.g., $0.99$--$0.9999$) often hurts exploration efficiency. Our interpretation is that overly large decay causes excessive additive superposition of co-activated features from different states, reducing the **identifiability / recoverability** of the underlying visitation pattern; whitening can mitigate, but not fully undo, this loss. We will add a sensitivity discussion in the revision.
>
> ### **Q3 & Q4. How does FPVR compare with history-dependent intrinsic reward methods, and does it remain effective in continuous control?**
>
> Thank you for raising these important questions. Mechanistically, FPVR differs from episodic-memory / hindsight-style intrinsic reward methods in how the history signal is used. Those methods compute a non-stationary intrinsic reward from episode history and rely on RL updates to propagate it. By contrast, Proposition 3.3 shows that FPVR admits an equivalent intrinsic-reward interpretation, but uses SR/SF to propagate this time-varying exploration signal immediately. More importantly, as shown in Figures 4, 5, and Appendix D.1, a well-learned SR can also provide a strong initial exploration prior under different reward distributions.
>
> To address both questions empirically, we added an additional **PointMaze** experiment, which provides an initial continuous-state-action evaluation and includes comparison with **ETD** [1]. The results further support the competitiveness of FPVR; see the anonymous supplementary results here: [PointMaze FPVR results](https://anonymous.4open.science/r/point_maze_fpvr-95EC). Since the current paper’s method is **not directly applicable to continuous action spaces**, this experiment uses our actor-critic/SAC-type FPVR extension; details are given in our response to **Reviewer dsZX, Q3**. We will include these results in the revision.
>
> ## **2. Weaknesses**
> ### **W1. Theoretical guarantees in continuous/high-dimensional spaces**
>
> Thank you for highlighting this important limitation. Please see **Q1**.
>
> ### **W2. Hyperparameter sensitivity**
>
> Thank you for pointing out this concern. Please see **Q2**.
>
> ### **W3 & W4. Lack of continuous-control / robotics evaluation and missing comparison with episodic-memory / hindsight intrinsic reward methods**
>
> Thank you for highlighting these important issues. Please see **Q3 & Q4**. To address both concerns, we added an additional **PointMaze** experiment and included comparison with **ETD** [1]. We will include these results in the revision.
>
> ### **Reference**
>
> [1] Y. Jiang *et al*., “Episodic Novelty Through Temporal Distance,” in *ICLR*, 2025.

---

> > ### Author Rebuttal · Reviewer_LNQk · 2026-04-02
> >
> > Thank you for the detailed rebuttal. I appreciate the additional clarifications and the effort to strengthen the paper. That said, I still have several deeper concerns that are only partially addressed. First, the method implicitly treats short-term future–past overlap as a proxy for exploratory redundancy, but it remains unclear when this surrogate is aligned with genuinely useful exploration rather than simply discouraging revisitation. In many long-horizon settings, revisiting bottleneck or decision-critical states can be necessary. Second, while the paper strongly motivates coverage efficiency, it does not fully justify when faster coverage translates into more reward-relevant exploration, especially in environments with substantial task-irrelevant variation. Third, FPVR is fundamentally representation-dependent: in practice it penalizes overlap in feature space rather than true state-visitation space, and the paper does not analyze how representation error, aliasing, or imperfect whitening may distort this signal. These concerns also make me uncertain about scalability to more realistic domains such as robot control, where the state space is high-dimensional, observations often contain large amounts of irrelevant variation, and effective exploration frequently requires structured revisitation rather than uniformly suppressing overlap. Overall, while the rebuttal improves the positioning of the paper, these concerns remain, so I will maintain my original score.

---

> > > ### Author Response · Authors · 2026-04-05
> > >
> > > Dear Reviewer LNQk,
> > >
> > > Thank you very much for your careful follow-up and for raising these deeper concerns. We sincerely appreciate the time and thought you have invested in understanding the method and in pushing us to clarify its scope and limitations more precisely.
> > >
> > > Regarding your first concern, we would like to clarify that FPVR is designed to suppress **short-term revisitation at a controllable timescale**, rather than to prevent revisitation in general. In FPVR, the effective suppression horizon is determined by the decay factors in the persistence and successor components: when these decay factors are smaller, the method only penalizes very recent overlap, whereas values closer to 1 extend the suppression window over a longer timescale. In our experiments, we typically find moderate values (roughly in the range 0.5--0.9) to work better than values extremely close to 1. Therefore, FPVR does not imply that an agent cannot revisit bottleneck or decision-critical states; rather, it discourages *immediate redundant return* to recently covered regions. We fully agree that some tasks may require more structured revisitation than FPVR encourages. In that sense, FPVR has its own domain of applicability, just as other exploration methods do. For example, RND [1] is well suited to broad novelty seeking in large state spaces, but, as our experiments suggest, it can be less effective in tasks where high coverage efficiency is especially important.
> > >
> > > Regarding your second concern, we agree that faster coverage does not automatically imply better reward-relevant exploration in every environment. However, improving coverage efficiency as a route to improving downstream learning is a common and well-established strategy in exploration and hierarchical RL. This is the central idea behind covering-option methods and related lines of work, including covering options [2], deep covering options [3], eigenoptions [4], and deep Laplacian-based options [5]. These methods differ in machinery and assumptions, but they share the view that better traversal of the state space can improve downstream exploration and learning. Our position is therefore not that coverage is universally sufficient, but that it is a principled and effective exploration bias in many sparse-reward and long-horizon settings, and FPVR is intended as a lightweight mechanism within this broader family.
> > >
> > > Regarding your third concern, we agree that the representation dependence of FPVR in high-dimensional settings deserves deeper analysis. In the current paper, whitening is used to mitigate feature co-activation and scale/correlation effects, but we do not claim that this fully resolves issues such as aliasing or representation error. We therefore agree that this remains an important limitation of the present manuscript. We are currently extending our study toward continuous-control navigation settings, which are closer to the downstream application scenarios we ultimately care about, such as robot navigation and manipulation. However, these experiments were originally planned as part of follow-up work rather than the scope of the present submission, so we have not yet included them in the paper. If you feel that seeing such preliminary navigation results would be helpful for your assessment of the method, we would be very happy to provide them through a follow-up update of this ''Reply Rebuttal Comment''.
> > >
> > > Thank you again for your careful reading, constructive skepticism, and many insightful questions. They have been very valuable in helping us understand which parts of the paper need clearer scope, stronger justification, and more explicit discussion of limitations.
> > >
> > > Wishing you all the best!
> > >
> > > ### References
> > >
> > > [1] Burda, Y., Edwards, H., Storkey, A., & Klimov, O. (2019). *Exploration by Random Network Distillation*. ICLR.
> > > [2] Jinnai, Y., Park, J. W., Abel, D., & Konidaris, G. (2019). *Discovering Options for Exploration by Minimizing Cover Time*. ICML.
> > > [3] Jinnai, Y., Park, J. W., Machado, M. C., & Konidaris, G. (2020). *Exploration in Reinforcement Learning with Deep Covering Options*. ICLR.
> > > [4] Machado, M. C., Bellemare, M. G., & Bowling, M. (2017). *A Laplacian Framework for Option Discovery in Reinforcement Learning*. ICML.
> > > [5] Klissarov, M., & Machado, M. C. (2023). *Deep Laplacian-based Options for Temporally-Extended Exploration*. ICML.

---

### Official Review · Reviewer_9kc2 · 2026-03-13

**Soundness:** 2
**Presentation:** 3
**Significance:** 2
**Originality:** 3
**Overall Recommendation:** 4
**Confidence:** 3

**Summary:**

This paper addresses the challenge of balancing state novelty with coverage efficiency in RL exploration. the author propose FPVR, an exploration mechanism induced by a Dual-Timescale State Representation that quantifies the overlap between predicted future occupancy and recorded past visitation. By applying FPVR as a bias to the Q-function, the method suppresses short-term revisitation to accelerate state-space coverage.

**Compliance With Llm Reviewing Policy:**

Affirmed.

**Final Justification:**

The authors' rebuttal successfully addressed my primary concerns regarding the theoretical risks of Q-function biasing and the reliability of Successor Representations (SR) during early exploration. Regarding the mathematical discrepancy in the recursive whitening implementation (Eq. 31), I accept the authors' explanation that this is a practical engineering approximation, provided it is clearly documented in the final manuscript.
Furthermore, the authors significantly strengthened the empirical evaluation by providing a comparison with the recent ETD baseline (Jiang et al., ICLR 2025), which helps situating the work within the current state-of-the-art landscape. While FPVR demonstrates a clear advantage in coverage speed and average adaptation time, I noted that it exhibits higher variance compared to traditional intrinsic reward baselines. I consider this a reasonable trade-off between exploration speed and stability.

**Key Questions For Authors:**

1.Could the authors clarify whether the bias injected into the Q-function decays over time or how the algorithm ensures convergence to the true optimal policy $\pi^*$?

2.The effectiveness of FPVR heavily relies on the Successor Representation (SR). In early exploration stages, the SR is often inaccurate due to the stochastic nature of the agent's policy. Could the authors discuss how the "future-past redundancy" remains a reliable signal when the "future" (SR) itself is poorly estimated?

3.In Eq. 31, the Persistence Feature (PF) is updated recursively using the whitening statistics $(\mu_t, \Sigma_t)$ from different time steps. Since these statistics change over time, does adding features processed by different whitening matrices lead to a "direction drift" or noise in the PF vector? How does this affect the accuracy of the cosine similarity?

4.In Eq.32, why is the bootstrap action $\bar{a}$ chosen by minimizing the redundancy $\tilde{\mathcal{F}}$ instead of following the agent's current policy or the greedy action? $\mathcal{F}$ itself is calculated using the SF. Does this circular dependency cause any training instability or lead to biased future prediction, especially in the early stages of learning?

5.Could the authors explain why the "timely update" advantage of FPCR does not translate into superior episodic return? In Figure 5, why does the intrinsic reward baseline $r^{FP}+SARSA$ outperform FPVR?

**Limitations:**

* Injecting a non-stationary bias directly into the Q-function potentially violates the Bellman optimality principle, and the authors do not provide evidence that the policy will eventually converge to the true optimal $\pi^*$ instead of a "novelty-biased" sub-optimal policy.
* The experimental evaluation is limited to older baselines (prior to 2024). Given the rapid progress in RL exploration, the exclusion of state-of-the-art methods from 2024 and 2025 makes it difficult to assess the actual competitiveness of FPVR.

**Strengths And Weaknesses:**

**Strengths:**
* The key idea of suppressing short-term revisitation is both intuitive and strategically sound. By explicitly penalizing the overlap between predicted future and recorded past, the method naturally encourages rapid state-space coverage.
* The proposed method is compatible with standard value-based and actor-critic algorithms. The incorcopration of online ZCA whitening demonstrates a sophisticated handling of feature correlations in high-dimensional state spaces.

**Weaknesses:**
* This paper lacks a comprehensive review of recent advancement in RL exploration, particularly missing key works from mid-2024 through 2025. Consequently, the empirical evaluation is limited by its reliance on outdated baselines, such as RND (2019), SR (2020), and SP (2023). While these methods are conceptually relevant to successor representations, they no longer represent the current SOTA in the rapidly evolving field of exploration in RL.
Without comparing FPVR against more recent, competitive baselines, it is difficult to assess its actual contribution. I recommend the authors consult curated resources,  the  ['Awesome Exploration Methods in RL'](https://www.google.com/url?sa=E&q=https%3A%2F%2Fgithub.com%2Fopendilab%2Fawesome-exploration-rl) repository, to identify and incorporate more modern benchmarks from top-tier conferences (2024–2025) into their evaluation.

* The authors claim that injecting FPVR as a bias into the Q-function is superior to intrinsic reward shaping. However, modiying the Q-value directly without theoretical guarantees may violate the Bellman optimality principle, potentially leading to sub-optimal policies.

---

> ### Author Rebuttal · Authors · 2026-03-30
>
> Dear Reviewer 9kc2,
>
> Thank you for your comprehensive feedback. Below we respond to the key questions and weaknesses.
>
> ## **1. Key Questions**
>
> ### **Q1. Optimality of Q-bias**
>
> Thank you for raising this important point. FPVR is not an arbitrary perturbation of Q. By Proposition 3.3, its unnormalized form is equivalent to the action value induced by a history-dependent intrinsic reward, with the same intuition extended to function approximation. Thus, FPVR shares the same conceptual basis as intrinsic-reward exploration.
>
> For off-policy methods, when FPVR is used only for action selection while the task critic/Q-function is still trained with the original extrinsic-reward target, FPVR changes the **behavior policy**, not the Bellman target. Hence it does not change the direction of the task objective, although an overly large bias can slow convergence.
>
> ### **Q2. Reliability of FPVR when SR is inaccurate in early exploration**
>
> Thank you for this thoughtful question. FPVR does not require accurate SR/SF; it relies on relative action differences, i.e., which action is more likely to return to recently visited regions.
>
> A tabular example illustrates this. Suppose SR is zero-initialized. Once a state $s_v$ has been visited, its PR entry satisfies $c_t(s_v)>0$. After a few SR updates, actions leading back to visited states will have $m(s,a)[s_v]>0$, so their overlap $\langle m(s,a),c_t\rangle$ and thus FPVR become positive. By contrast, actions leading toward unvisited regions initially have $0$ overlap. Therefore, even a crude SR biases FPVR against recently visited states; the same intuition carries over to function approximation.
>
> Empirically, Appendix D.1 shows that a **well-learned SR can further improve** exploration efficiency, but this does not mean FPVR is ineffective while SR is still being learned. On the contrary, Figures 4, 5, and 6 already show that FPVR, trained online from scratch, achieves higher early exploration efficiency than the baselines.
>
> ### **Q3. Whitening drift**
>
> Thank you for highlighting this issue. At time $t$, FPVR is computed under the current whitening statistics $(\mu_t,\Sigma_t)$. Thus, both PF and SF are expressed in the **same whitening-conditioned coordinate** system, so FPVR does not compare vectors from mismatched spaces. Instead, whitening defines a unified time-varying similarity metric induced by recent state statistics, analogous to a Mahalanobis-type cosine similarity. We will clarify this in the revision.
>
> ### **Q4. Min-redundancy bootstrap**
>
> Thank you for raising this important question. We use the min-redundancy bootstrap action because the SR/SF in FPVR is intended to support the exploration objective, so this target naturally aligns learning with minimizing future-past overlap. This is also analogous to the off-policy idea in Q-learning/DQN, where the bootstrap target need not match the current behavior policy. Importantly, this choice is **not critical**: Appendix D.2 shows that `current`, `min`, and `mean` targets achieve similar overall coverage, with `min` and `mean` being slightly better in the visual setting.
>
> ### **Q5. Why does the “timely update” advantage not always translate into higher episodic return?**
>
> Thank you for raising this question. We apologize that the original presentation was not sufficiently intuitive in showing FPVR’s faster adaptation. Although Figure 5 makes FPVR and its equivalent intrinsic-reward version appear visually close, a closer comparison still favors FPVR: across the 5 reward-reset events, FPVR reaches return 1 first in 2 cases, while $r^{FP}$ does so in only 1 case; their average ranks are 2.0 and 2.4, respectively.
>
> To make this comparison more direct, under the same setting as Figure 5, we increased the number of reward changes to 100 and measured the steps after each reset until the episode return first reached 1 (or 40000 if it never did):
>
> | Method | Mean steps to first reach return 1 ↓ | Std. |
> |---|---:|---:|
> | **FPVR+SARSA** | **9718.8** | 3855.9 |
> | $r^{FP}$+SARSA | 11582.2 | 3075.7 |
> | SP+SARSA | 12780.1 | 4118.1 |
> | SR+SARSA | 20247.7 | 14616.3 |
> | SARSA | 31732.9 | 23148.5 |
>
> We will add this result in the revision.
>
> ## **2. Weaknesses**
>
> ### **W1. Limited comparison to recent methods**
>
> Thank you for highlighting this concern. Our original baselines were chosen mainly for mechanistic relevance to FPVR and consistency with our environments, rather than recency alone.
>
> To address this, we added a comparison with **ETD** [1], a recent episodic intrinsic exploration method. The additional PointMaze results further support FPVR; see the [anonymous supplementary link](https://anonymous.4open.science/r/point_maze_fpvr-95EC). We will include these results in the revision.
>
> ### **W2. Non-stationary Q-bias**
>
> Thank you for your comments. Please refer to our response to **Q1**.
>
> Wish you all the best!
>
> ### **Reference**
> [1] Y. Jiang *et al*., “Episodic Novelty Through Temporal Distance,” in *ICLR*, 2025.

---

> > ### Author Rebuttal · Reviewer_9kc2 · 2026-04-03
> >
> > I thank the authors for their detailed rebuttal. Regarding Q3, while the recursive implementation of the Persistence Feature (Eq. 31) remains a mathematical approximation of the formal definition due to shifting whitening statistics, it appears acceptable for practical implementation. I recommend the authors explicitly clarify this approximation and its potential implications in the revised manuscript.
> >
> > Furthermore, regarding the adaptation performance in Figure 5, the authors' additional analysis indicates that while FPVR+SARSA leads the $r^{FP}+SARSA$ baseline on average, it also exhibits a notably larger standard deviation. This suggests that the 'timely update' advantage may come at the cost of higher variance or reduced stability in certain scenarios. Given these clarifications and the supplementary data provided, I am increasing my score, though I will adjust my confidence level accordingly.

---

> > > ### Author Response · Authors · 2026-04-04
> > >
> > > Dear Reviewer 9kc2,
> > >
> > > Thank you very much for your follow-up and for your generous response to our rebuttal. We are truly encouraged to know that our clarifications and additional analysis have addressed your concerns.
> > >
> > > We also greatly appreciate your further suggestions for the revised manuscript. In particular, we will explicitly clarify that the recursive implementation of the Persistence Feature in Eq. 31 is a practical approximation under shifting whitening statistics, and we will also note that the faster adaptation of FPVR may be accompanied by higher variance in some scenarios.
> > >
> > > Your feedback has been very helpful in improving both the presentation and the interpretation of our method. Should you choose to reflect this updated assessment in your score, we would sincerely appreciate it (we understand that any score update would need to be submitted through the final justification before 4.7 AOE). Thank you again for the time and effort you have devoted to our paper.
> > >
> > > Wishing you all the best!

---

### Decision · Program_Chairs · 2026-04-30

**Decision:**

Reject

**Comment:**

While the premise of minimizing the similarity between future and past state visitations is conceptually intriguing, the manuscript suffers from a critical theoretical flaw. Specifically, the formulation of $M(s, a, s^\prime)$ and the associated metric $\mathbf{m}(s, a)$ are intrinsically dependent on the policy, yet they are concurrently used to update that very same policy. This introduces a fundamental circular dependency within the proposed algorithm. Furthermore, as correctly noted by Reviewer 9kc2, the methodological design is fundamentally indirect. A mathematically sound and far more intuitive approach would be to directly optimize the policy to minimize the proposed criterion, rather than artificially modifying the Q-function to force a policy change. Because this circularity fundamentally undermines the theoretical validity of the method and obscures its core contributions, I recommend rejection.